# Elucidating the Design Space of Decay in Linear Attention

[1]**Zhen Qin,** [2]**Xuyang Shen,** [2]**Yiran Zhong**[*]
[1]TapTap [2]OpenNLPLab

 https://github.com/Doraemonzzz/xmixers

## Abstract

This paper presents a comprehensive investigation into the decay mechanisms inherent in linear complexity sequence models. We systematically delineate the design space of decay mechanisms across four pivotal dimensions: *parameterization strategy*, which refers to the computational methodology for decay; *parameter sharing*, which involves the utilization of supplementary parameters for decay computation; *decay granularity*, comparing scalar versus vector-based decay; and *compatibility with relative positional encoding methods*, such as Rotary Position Embedding (RoPE). Through an extensive series of experiments conducted on diverse language modeling tasks, we uncovered several critical insights. Firstly, the design of the parameterization strategy for decay requires meticulous consideration. Our findings indicate that effective configurations are typically confined to a specific range of parameters. Secondly, parameter sharing cannot be used arbitrarily, as it may cause decay values to be too large or too small, thereby significantly impacting performance. Thirdly, under identical parameterization strategies, scalar decay generally underperforms compared to its vector-based counterpart. However, in certain scenarios with alternative parameterization strategies, scalar decay may unexpectedly surpass vector decay in efficacy. Lastly, our analysis reveals that RoPE, a commonly employed relative positional encoding method, typically fails to provide tangible benefits to the majority of linear attention mechanisms.

## 1 Introduction

Linear complexity sequence models have recently gained prominence as a viable alternative to Transformer models, primarily due to their ability to circumvent the latter's inherent quadratic computational complexity. This category of models encompasses a diverse range of architectures, including linear recurrent neural networks (Peng et al., 2023a; Orvieto et al., 2023; Qin et al., 2023d; De et al., 2024; Orvieto et al., 2023), state space models (Gu et al., 2022a; Smith et al., 2023; Gu & Dao, 2023; Dao & Gu, 2024), and various formulations of linear attention mechanisms (Sun et al., 2023; Qin et al., 2023b; Yang et al., 2023; 2024; Arora et al., 2024). It is noteworthy that these methods can be unified within the framework of linear attention (Chou et al., 2024; Yang et al., 2023). Therefore, we employ linear attention to represent these methods in the subsequent section. Within this domain, decay mechanisms (Qin et al., 2024a) have been instrumental in bolstering the modeling accuracy of these linear models. By selectively emphasizing pertinent contextual information and concurrently diminishing the influence of less significant historical signals, these mechanisms optimize the allocation of computational resources and representational capacity, thereby enhancing overall model performance.

Despite the pivotal role of the decay component in determining the efficacy of linear attention mechanisms, the architectural choices that govern its implementation have not yet been subjected to rigorous and systematic evaluation. The landscape of linear attention variants is characterized by significant heterogeneity in decay design, encompassing a wide range of approaches such as constant decay coefficients (Qin et al., 2024a; Sun et al., 2023),

---

[*] Corresponding author. Email: zhongyiran@gmail.com.

context-sensitive formulations (Peng et al., 2021), and varying levels of granularity, as exemplified by scalar (Dao & Gu, 2024) and vectorized (Gu & Dao, 2023) implementations. These diverse methodologies have typically been proposed in isolation, without the benefit of controlled comparative analyses that could elucidate their relative strengths and limitations. Such a lack of comprehensive evaluation hinders a deeper understanding of the trade-offs inherent in different decay mechanisms and impedes the identification of optimal design choices for specific application scenarios.

In this paper, we conduct a comprehensive investigation of the design space of decay mechanism in linear attention. We conceptualize this design space along the following four fundamental dimensions:

- **Parameterization strategy**: The algorithmic approach for computing decay values, encompassing static, trainable, and input-conditional formulations.
- **Parameter sharing**: The architectural decision regarding whether to allocate dedicated parameters specifically for decay computation.
- **Decay granularity**: The structural choice between uniform scalar decay across all dimensions versus fine-grained vector decay with dimension-specific coefficients.
- **Positional encoding integration**: The interaction patterns between decay mechanisms and positional information, with a particular focus on compatibility with RoPE-based (Su et al., 2021) encoding strategies.

Through a comprehensive empirical evaluation conducted on language modeling benchmarks using the fineweb-edu-10b dataset (Penedo et al., 2024), we systematically assess the design variations across these dimensions. Our investigation reveals several significant findings: the parameterization strategy exhibits notable sensitivity, with effective configurations predominantly clustering within specific parametric regions; parameter sharing may lead to decay values close to 0 or 1, thereby significantly affecting performance; the relative performance of scalar versus vector decay mechanisms is critically contingent upon the underlying parameterization approach; and, RoPE positional encodings generally do not enhance performance across the majority of linear attention.

This systematic exploration illuminates previously obscured relationships between architectural choices in linear attention design. By mapping this design space comprehensively, our work provides researchers and practitioners with actionable insights for developing more efficient and effective attention mechanisms. These findings establish a foundation for principled design decisions when implementing linear attention variants across diverse applications and computational environments.

## 2 Related work

### 2.1 Linear Complexity Sequence Model

**Linear Attention** transforms the conventional softmax attention mechanism to attain linear computational complexity relative to sequence length. This is achieved by leveraging the kernel trick (Katharopoulos et al., 2020) to decompose attention computation into inner products of hidden representations, thereby circumventing the need for softmax calculations. Various implementations employ distinct kernel functions, such as the '1+elu' function by Katharopoulos et al. (2020), cosine functions by Qin et al. (2021), and theoretical approximation by Choromanski et al. (2021); Peng et al. (2021). Despite these innovations, early implementations often lagged behind standard Transformers due to **attention dilution** (Qin et al., 2022). Subsequent research by Qin et al. (2024a) and Sun et al. (2023) demonstrated that integrating suitable decay mechanisms significantly bolsters the representational capacity of linear attention, enabling performance comparable to or approaching that of standard softmax attention. Further advancements by Yang et al. (2023); Peng et al. (2024) enhanced model performance through the introduction of data dependency.

**State Space Models** elegantly reformulate sequence modeling within a continuous-time dynamical systems framework (Gu et al., 2020; 2022a). They perform sequence modeling by discretizing state space equations in continuous space (Gu et al., 2020; 2022a), improve training stability through careful initialization (Gu et al., 2023), simplify the model through diagonalization assumption (Gupta et al., 2022), and enhance model performance through data dependency (Gu & Dao, 2023; Dao & Gu, 2024). Their theoretical foundation enables

computational efficiency while maintaining powerful expressivity, providing a mathematically rigorous modeling paradigm for long sequence processing.

**Linear Recurrent Neural Networks** (Martin & Cundy, 2018) enables parallel computation by removing the nonlinear dependencies of traditional RNNs (Chung et al., 2014; Hochreiter & Schmidhuber, 1997). These models ingeniously utilize linear recursive structures to effectively capture long-distance dependencies without global attention computations. Representative implementations such as Hgrn1 (Qin et al., 2023d) and RWKV-4 (Peng et al., 2023b) demonstrate capabilities comparable to similarly-scaled Transformer architectures through carefully designed structures and linear complexity operations. Hgrn2 (Qin et al., 2024b) further enhances the capability of Linear RNN through state expansion and establishes the connection between Linear RNN and Linear Attention.

### 2.2 Relative Positional Encoding

Relative positional encodings are widely used in Transformers; however, most of them are incompatible with Linear Attention because they typically require computing the Attention Matrix, i.e., $\mathbf{Q}\mathbf{K}^\top$, which is not permitted in Linear Attention. LRPE (Qin et al., 2023c) points out that a prerequisite for compatibility between relative positional encoding and Linear Attention is decomposability, meaning that relative position information can be captured by separately operating on $\mathbf{Q}$ and $\mathbf{K}$. Common examples include RoPE (Su et al., 2021) and LRPE. The former uses rotary positional encoding to capture relative position information:

$$\mathbf{x}_t^j = \mathbf{R}_t \mathbf{x}_t^j \in \mathbb{R}^{d/h}, \mathbf{R}_t = \mathrm{diag}(\mathbf{R}_{t,1}, \ldots, \mathbf{R}_{t,d/2}) \in \mathbb{R}^{d/h \times d/h}, \mathbf{R}_{t,k} = \begin{bmatrix} \cos(t\theta_k) & -\sin(t\theta_k) \\ \sin(t\theta_k) & \cos(t\theta_k) \end{bmatrix}, \mathbf{x} \in \{\mathbf{q}, \mathbf{k}\}.$$

The latter employs a Cosine reweighting mechanism to capture relative position information:

$$\mathbf{x}_t^j = \mathrm{concat}[\mathbf{x}_t^j \cos(t\theta^j), \mathbf{x}_t^j \sin(t\theta^j)] \in \mathbb{R}^{2d/h}, \theta^j \in \mathbb{R}^{d/h}, \mathbf{x} \in \{\mathbf{q}, \mathbf{k}\}.$$

Another relative positional encoding, TPE, was proposed in (Qin et al., 2025), which differs from previous approaches in that it operates after the embedding layer rather than in the attention layer, and it only operates once. It uses Toeplitz matrices (Qin et al., 2023a) to capture relative position information and is parameterized by SSM (Gu et al., 2022b; Ma et al., 2022; 2024), in the form:

$$\mathbf{o}_t^j = \sum_{j=1}^t r_{t-j}^j \mathbf{x}_j^j = \sum_{j=1}^t (\mathbf{a}^j)^\top (\mathbf{b}^j)(\lambda^j)^{t-j} \mathbf{x}_j^j.$$

## 3 Preliminary

Yang et al. (2023; 2024); Chou et al. (2024) points out that the aforementioned methods can be unified under linear attention mechanisms, with the general mathematical formulation:

$$\mathbf{s}_t^j = \mathbf{M}_t^j \mathbf{s}_{t-1}^j + \mathbf{k}_t^j (\mathbf{v}_t^j)^\top, (\mathbf{o}_t^j)^\top = (\mathbf{q}_t^j)^\top \mathbf{s}_t^j, t = 1, \ldots, n. \tag{1}$$

where $n$ represents sequence length, $\mathbf{q}_t^j, \mathbf{k}_t^j \in \mathbb{R}^{d/h}, \mathbf{v}_t^j \in \mathbb{R}^{e/h}$ correspond to the query, key, and value vectors at position $t$ for head $j$, $h$ denotes the number of heads, $d, e$ denotes of query/key hidden dimension and value hidden dimension, $\mathbf{s}_t^j \in \mathbb{R}^{d/h \times e/h}$ denotes the state matrix of linear attention, $\mathbf{o}_t^j \in \mathbb{R}^e$ is the output vector, and $\mathbf{M}_t^j \in \mathbb{R}^{d/h \times d/h}$ represents the state transition matrix. Generally, $\mathbf{y}_t^j = f_{\mathbf{y}}(\mathbf{x}_t^j), \mathbf{y} \in \{\mathbf{q}, \mathbf{k}, \mathbf{v}, \mathbf{M}\}$, where $\mathbf{x}_t^j$ is the input representation at position $t$ for head $j$, and $f_y$ is a function mapping, indicating that $\mathbf{y}_t^j$ has **data dependency** on input $\mathbf{x}_t^j$.

The state transition matrix $\mathbf{M}_t^j$ typically adopts two characteristic structures: a diagonal matrix form $\mathrm{diag}(\lambda_t^j)$ (Yang et al., 2023; Qin et al., 2024b; Zhang et al., 2024; Beck et al., 2024), or a DPLR (diagonal plus low rank) structure $\mathrm{diag}(\lambda_t^j) + \mathbf{a}_t^j (\mathbf{b}_t^j)^\top$, where $\lambda_t^j, \mathbf{a}_t^j, \mathbf{b}_t^j \in \mathbb{R}^{d/h}$ (Yang et al., 2025; Peng et al., 2025; Gu et al., 2022a). Our paper focuses on the

Table 1: Taxonomy of decay mechanisms in various linear attention variants. Different implementations exhibit unique parametrization strategies and structural characteristics. Here, $j$ denotes the head index (out of $h$ total heads), and $l$ represents the layer index (out of $L$ total layers). $\mathbf{A}^j, \Delta^j_t, \tau \in \mathbb{R}$, and $\mathbf{f}^j_t \in \mathbb{R}^{d/h}$ for vector decay or $\mathbf{f}^j_t \in \mathbb{R}$ for scalar decay. lse represents the logsumexp operator, i.e., $\mathrm{lse}(\mathbf{x}) = \log \sum \exp(x_i)$, and $\mathrm{sigmoid}(\mathbf{x}) = 1/(1 + \exp(-\mathbf{x}))$.

| Method | Parameterization Strategy | Parameter Sharing | Scalar | Recurrence Formula |
|---|---|---|---|---|
| Mamba2 | $\lambda^j_t = \mathrm{sigmoid}\left(-\mathbf{f}^j_t - \Delta^j\right)^{\exp(\mathbf{A}^j)}$ | ✗ | ✓ | |
| Mamba2 wo $\mathbf{A}$ | $\lambda^j_t = \mathrm{sigmoid}\left(-\mathbf{f}^j_t - \Delta^j\right)$ | ✗ | ✓ | |
| Mamba2 wo $\Delta$ | $\lambda^j_t = \mathrm{sigmoid}\left(-\mathbf{f}^j_t\right)^{\exp(\mathbf{A}^j)}$ | ✗ | ✓ | $\mathbf{s}^j_t = \lambda^j_t \mathbf{s}^j_{t-1} + \mathbf{k}^j_t(\mathbf{v}^j_t)^\top$ |
| Mamba2 wo $\mathbf{A}$ & $\Delta$ | $\lambda^j_t = \mathrm{sigmoid}\left(-\mathbf{f}^j_t\right)$ | ✗ | ✓ | |
| GLA | $\lambda^j_t = \mathrm{sigmoid}(\mathbf{f}^j_t)^{1/\tau}$ | ✗ | ✗ | $\mathbf{s}^j_t = \mathrm{diag}(\lambda^j_t)\mathbf{s}^j_{t-1} + \mathbf{k}^j_t(\mathbf{v}^j_t)^\top$ |
| Hgrn2 | $\lambda^j_t = \lambda^j + (1 - \lambda^j)\mathrm{sigmoid}(\mathbf{f}^j_t)$ | ✓ | ✗ | $\mathbf{s}^j_t = \mathrm{diag}(\lambda^j_t)\mathbf{s}^j_{t-1} + (1 - \lambda^j_t)(\mathbf{v}^j_t)^\top$ |
| Lightnet | $\lambda^j_t = \exp(\mathrm{lse}(\mathbf{f}^j_{<t-1}) - \mathrm{lse}(\mathbf{f}^j_{<t}))$ | ✓ | ✗ | $\mathbf{s}^j_t = \mathrm{diag}(\lambda^j_t)\mathbf{s}^j_{t-1} + (1 - \lambda^j_t)(\mathbf{v}^j_t)^\top$ |
| TNL | $\lambda^j_t = \exp\left(-8j/h \times (1 - l/L)\right)$ | ✗ | ✓ | $\mathbf{s}^j_t = \lambda^j \mathbf{s}^j_{t-1} + \mathbf{k}^j_t(\mathbf{v}^j_t)^\top$ |
| Simple Decay | $\lambda^j_t = \mathrm{sigmoid}\left(\mathbf{f}^j_t + \Delta^j\right)$ | ✗ | both | $\mathbf{s}^j_t = \mathrm{diag}(\lambda^j_t)\mathbf{s}^j_{t-1} + \mathbf{k}^j_t(\mathbf{v}^j_t)^\top$ |

diagonal component $\mathrm{diag}(\lambda^j_t)$, defining it as the decay mechanism (Qin et al., 2024a), and systematically explores its design space. In this case, the recurrence simplifies to:

$$\mathbf{s}^j_t = \mathrm{diag}(\lambda^j_t)\mathbf{s}^j_{t-1} + \mathbf{k}^j_t(\mathbf{v}^j_t)^\top, c \in \{0, 1\}. \tag{2}$$

Different model architectures employ diverse decay design strategies. For instance, TNL (Qin et al., 2024a) and RetNet (Sun et al., 2023) utilize data-independent scalar decay with predetermined fixed values. Mamba2 (Dao & Gu, 2024) introduces data-dependent scalar decay, computing dynamic decay values through discretization methods. GLA (Yang et al., 2023) adopts a vector decay strategy, calculating decay vectors using sigmoid functions with temperature parameters. Hgrn2 (Qin et al., 2024b) similarly implements vector decay, but innovatively shares decay parameters with key vectors and computes decay values through lower bound constraints and sigmoid functions. Table 1 summarizes the decay mechanism design taxonomy across various linear attention variants, establishing the foundation for our subsequent systematic investigation.

## 4 The Design Space of Decay

As illustrated in the previous section, it is evident that various methods utilize distinct parameterization schemes for computing decay. These schemes can significantly influence the effectiveness and efficiency of the decay mechanism within linear attention models. Additionally, there is a distinction in how parameters are handled between decay and key calculations, with some methods opting for **Parameter Sharing**, where the same parameters are used for both key and decay, while others employ independent parameters, allowing for more flexibility and potentially reducing interference between the two computations.

Moreover, the granularity of decay application varies across methods. Some approaches apply **Vector Decay**, where different decay values are assigned to each feature, enabling a more nuanced and feature-specific control over the decay process. In contrast, other methods implement **Scalar Decay**, where a uniform decay value is applied across all features for each head, simplifying the computation but potentially at the cost of expressiveness.

Furthermore, the parameterization schemes for scalar decay exhibit significant diversity. Some models integrate decay mechanisms with relative positional encoding (Qin et al., 2025), which raises the question of whether this integration is truly necessary or if it introduces unnecessary complexity.

Based on these observations, we propose a design space for decay mechanisms with the following dimensions:

**Parameterization strategy** : How to calculate decay values.

**Parameter sharing** : Whether to share parameters with other components.

**Decay granularity** : Using Vector Decay or Scalar Decay.

**Relative positional encoding** : Whether relative position encoding is needed.

To ensure a fair comparison, all methods use the same network architecture as shown in Figure 3 in the appendix, which we call the Decay Linear Transformer. Each Decay Linear Transformer consists of multiple Decay Linear Transformer Layers, with each layer comprising a Token Mixer and Channel Mixer. For the Channel Mixer, we employ GLU (Shazeer, 2020); for the Token Mixer, we implement Linear Attention with decay, creating different variants through different decay strategies implemented via FLA (Yang & Zhang, 2024a) and Xmixers (Qin, 2025). We uniformly use the silu function as the kernel function for query and key in Linear Attention. We adopt the low-rank sigmoid output gate and normalization strategy from TNL (Qin et al., 2024a), using RMSNorm for all normalization operations. For all low-rank projections, we consistently use an intermediate dimension of $d/h$ (Qin et al., 2024a). In subsequent discussions, we assume that $\mathbf{w}_k^\top$ represents the $k$-th row of matrix $\mathbf{W}$.

The computation for the Linear Attention can be expressed as follows:

$$\mathbf{s}_t^j = \mathrm{diag}(\lambda_t^j)\mathbf{s}_{t-1}^j + \mathbf{k}_t^j(\mathbf{v}_t^j)^\top, (\mathbf{o}_t^j)^\top = (\mathbf{q}_t^j)^\top \mathbf{s}_t^j, t = 1, \ldots, n. \tag{3}$$

Where:

$$\mathbf{Q}^j = g(\mathbf{X}\mathbf{W}_q^j), \mathbf{K}^j = g(\mathbf{X}\mathbf{W}_k^j), \mathbf{V}^j = \mathbf{X}\mathbf{W}_v^j, \mathbf{W}_y^j \in \mathbb{R}^{d \times d/h}, y \in \{q, k, v\}, g = \mathrm{silu}, j = 1, \ldots, h.$$

For decay $\lambda_t^j$, we first obtain activation $\mathbf{f}_t^j$ through linear layers, then calculate $\lambda_t^j = f(\mathbf{f}_t^j)$ through function $f$ (whose form is determined by the **Parameterization Strategy**). The detailed formulation can be found in Appendix A.2.

The final output of the Token Mixer layer is:

$$\mathbf{O} = \mathrm{Norm}(\mathrm{concat}([\mathbf{O}^1, \ldots, \mathbf{O}^h]) \odot \mathbf{U}), \mathbf{U} = \mathrm{sigmoid}(\mathbf{X}\mathbf{W}_{u_1}\mathbf{W}_{u_2}), \mathbf{W}_{u_1} \in \mathbb{R}^{d \times d/h}, \mathbf{W}_{u_2} \in \mathbb{R}^{d/h \times d}.$$

**Parameterization Strategy**    The parameterization strategy delineates the method by which decay values are computed. In our investigation, we examined a variety of parameterization schemes, conducting a comparative analysis of Mamba, GLA, Hgrn2, and LightNet within the context of Vector Decay. For the Scalar Decay scenario, we incorporated TNL and its learnable variant, Learnable TNL (TNL-L), where TNL-L utilizes the initialization of TNL but permits further learning. Additionally, we extended the application of Mamba Decay from a scalar to a vector context and conversely, adapted the decay mechanisms of GLA, Hgrn2, and LightNet from a vector to a scalar framework.

**Parameter Sharing**    The parameter-sharing dimension investigates whether supplementary parameters are designated for the computation of decay. Following the calculation of the decay term $\lambda_t^j$, we define the key component as $\mathbf{k}_t^j = 1 - \lambda_t^j$. In this particular analysis, we focused exclusively on the Vector Decay approach and conducted a comparative evaluation of Mamba2, GLA, Hgrn2, and LightNet.

**Decay Granularity**    Decay granularity pertains to the distinction between scalar decay, which applies a uniform decay value across all dimensions, and vector decay, which employs independent decay values for each individual dimension. In our study, we systematically compared the effects of scalar decay and vector decay for the models Mamba, GLA, Hgrn2, and LightNet, evaluating their respective impacts on performance and computational efficiency.

**Compatibility with Positional Encoding**    Additionally, we explored the compatibility of various decay mechanisms with relative positional encoding. For this investigation, we selected RoPE and TPE as the RPE candidates. We exclude LRPE due to its drawback of doubling the head dimension of the Query and Key vectors, which consequently increases the computational cost. We opted for Scalar Decay for this analysis because RoPE exhibits compatibility issues with Vector Decay. Naive implementation of Vector Decay with RoPE fails to accurately represent relative position information (see Appendix A.3 for details).

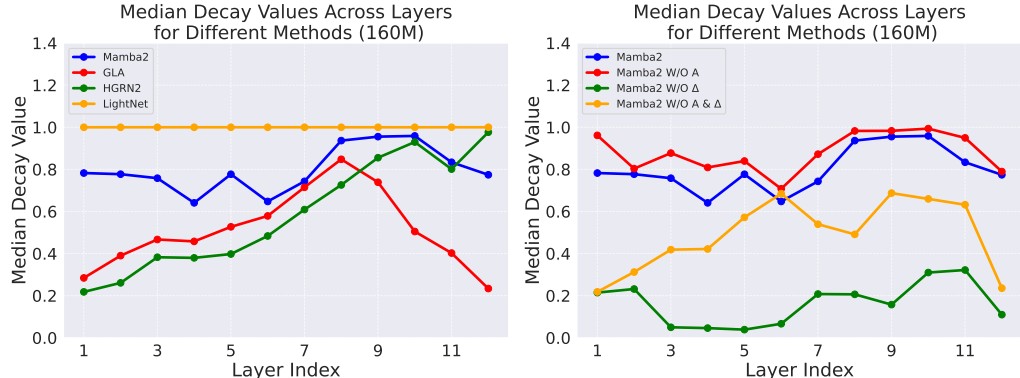

Figure 1: Distribution of median decay values for each layer across different methods, with model size of 160M. **Left figure**: Median distribution of Vector Decay. **Right figure**: Median distribution of Mamba ablation under Vector Decay.

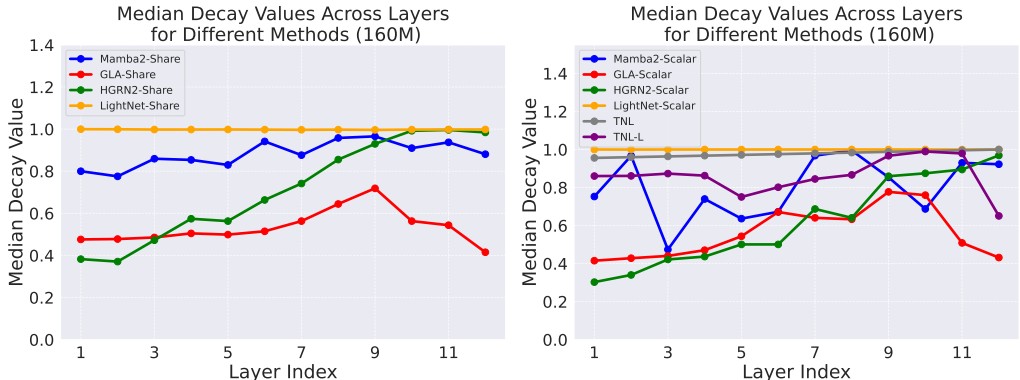

Figure 2: Distribution of median decay values for each layer across different methods, with model size of 160M. **Left figure**: Median distribution of Share Decay. **Right figure**: Median distribution of Scalar Decay.

## 5 Experiments

We conducted a series of language modeling experiments utilizing the fineweb-edu-10B dataset (Penedo et al., 2024). Our experiments involved training language models with varying parameter sizes, specifically 160 million, 410 million, and 1.4 billion parameters. The detailed configurations for these models are provided in Table 5. For tokenization, we employed the GPT2-Tokenizer (Radford et al., 2019).

The training process was governed by several key hyperparameters: a global batch size of 256, a sequence length of 2048, and the AdamW optimizer (Loshchilov & Hutter, 2019) with $\beta_1 = 0.9$ and $\beta_2 = 0.999$. The learning rate was set to $3 \times 10^{-4}$. We utilized the WSD scheduler (Hu et al., 2024) and trained the models for 20,000 steps. Our implementation was grounded in the Flame (Zhang et al., 2025), FLA (Yang & Zhang, 2024b), Xmixers (Qin, 2025), and PyTorch (Paszke et al., 2019) frameworks. All models were trained using 8 NVIDIA A100 GPUs. We evaluated the models using the lm-eval-harness (Gao et al., 2021) to perform zero-shot evaluation.

### 5.1 Parameterization Strategy

We conducted a comparative analysis of several parameterization strategies for Vector Decay, including Mamba2, GLA, Hgrn2, and LightNet. The results of this comparison are presented in Tables 2, 6, and 7. The findings indicate that Mamba2 exhibited superior performance across all model sizes, followed by Hgrn2, GLA, and LightNet, respectively.

Table 2: Performance comparison of different model methods under various configurations (1.45B parameters). AVG represents average perplexity (lower is better) or average correct score rate.

| Method | Pa | Loss | PPL ↓ | | | Accuracy ↑ | | | | | | | | |
|---|---|---|---|---|---|---|---|---|---|---|---|---|---|---|
| | | | Wiki | LMB | AVG | BOQA | PIQA | Hella | Wino | ARC-e | ARC-c | OBQA | SOQA | AVG |
| *Vector decay* | | | | | | | | | | | | | | |
| Mamba2 | 1.45 | 2.513 | 22.8 | 25.1 | 24.0 | 61.7 | 70.0 | 47.7 | 52.8 | 67.1 | 30.7 | 37.6 | 39.8 | 50.9 |
| GLA | 1.45 | 2.530 | 23.4 | 29.4 | 26.4 | 57.3 | 69.3 | 47.3 | 54.1 | 66.5 | 33.8 | 36.6 | 39.8 | 50.6 |
| Hgrn2 | 1.45 | 2.526 | 23.2 | 24.3 | 23.8 | 59.7 | 70.1 | 47.2 | 52.5 | 65.9 | 33.5 | 35.4 | 39.5 | 50.5 |
| LightNet | 1.45 | 2.561 | 25.2 | 34.8 | 30.0 | 58.9 | 69.4 | 43.3 | 53.8 | 64.6 | 30.6 | 34.8 | 40.1 | 49.4 |
| *Mamba ablation* | | | | | | | | | | | | | | |
| Mamba2 w/o **A** | 1.45 | 2.513 | 22.8 | 24.3 | 23.5 | 62.4 | 69.9 | 47.4 | 55.6 | 66.6 | 32.1 | 33.2 | 40.1 | 50.9 |
| Mamba2 w/o Δ | 1.45 | 2.585 | 25.3 | 31.5 | 28.4 | 58.5 | 68.9 | 44.7 | 50.8 | 64.9 | 30.2 | 36.4 | 39.8 | 49.3 |
| Mamba2 w/o **A**,Δ | 1.45 | 2.526 | 23.4 | 25.8 | 24.6 | 61.0 | 69.6 | 47.7 | 53.2 | 67.5 | 32.4 | 38.0 | 39.5 | 51.1 |
| *Parameter share* | | | | | | | | | | | | | | |
| Mamba2 | 1.45 | 2.517 | 22.8 | 24.6 | 23.7 | 60.7 | 69.9 | 47.4 | 54.1 | 66.8 | 30.6 | 36.8 | 39.9 | 50.8 |
| GLA | 1.45 | 2.583 | 25.5 | 35.9 | 30.7 | 61.7 | 69.4 | 45.5 | 50.8 | 65.5 | 30.9 | 35.0 | 39.4 | 49.8 |
| Hgrn2 | 1.45 | 2.529 | 23.3 | 24.2 | 23.7 | 58.0 | 70.2 | 47.2 | 51.1 | 67.0 | 31.2 | 36.2 | 40.2 | 50.1 |
| LightNet | 1.45 | 2.620 | 26.0 | 49.1 | 37.6 | 60.9 | 68.8 | 42.7 | 50.9 | 61.5 | 30.5 | 33.8 | 38.8 | 48.5 |
| *Scalar decay* | | | | | | | | | | | | | | |
| Mamba2 | 1.45 | 2.529 | 23.4 | 28.3 | 25.8 | 56.6 | 69.3 | 47.0 | 51.7 | 66.7 | 31.7 | 38.2 | 40.9 | 50.3 |
| GLA | 1.45 | 2.550 | 23.8 | 28.9 | 26.3 | 60.6 | 70.0 | 46.3 | 52.6 | 65.9 | 32.7 | 35.8 | 40.1 | 50.5 |
| Hgrn2 | 1.45 | 2.541 | 24.2 | 32.0 | 28.1 | 60.0 | 69.3 | 45.9 | 53.5 | 66.0 | 30.7 | 35.0 | 39.4 | 50.0 |
| LightNet | 1.45 | 2.574 | 24.3 | 33.3 | 28.8 | 62.0 | 69.3 | 45.1 | 51.3 | 65.3 | 29.7 | 36.0 | 38.7 | 49.7 |
| TNL | 1.45 | 2.552 | 24.3 | 29.4 | 26.9 | 61.3 | 69.9 | 45.9 | 53.8 | 66.6 | 30.3 | 34.8 | 40.3 | 50.4 |
| TNL-L | 1.45 | 2.545 | 23.7 | 29.0 | 26.4 | 59.6 | 70.7 | 46.1 | 51.4 | 64.1 | 30.0 | 35.8 | 39.3 | 49.6 |
| *Rope* | | | | | | | | | | | | | | |
| Mamba2 | 1.45 | 2.531 | 23.5 | 28.2 | 25.9 | 60.7 | 69.4 | 46.6 | 53.7 | 65.7 | 30.9 | 35.6 | 40.3 | 50.4 |
| GLA | 1.45 | 2.580 | 25.5 | 35.0 | 30.2 | 60.1 | 69.0 | 45.3 | 54.2 | 65.2 | 31.6 | 35.4 | 39.1 | 50.0 |
| Hgrn2 | 1.45 | 2.560 | 24.6 | 29.3 | 27.0 | 59.1 | 69.2 | 45.6 | 51.5 | 66.0 | 31.7 | 35.4 | 39.9 | 49.8 |
| LightNet | 1.45 | 2.570 | 24.5 | 30.1 | 27.3 | 61.4 | 69.4 | 45.5 | 52.4 | 64.9 | 29.5 | 34.6 | 39.1 | 49.6 |
| TNL | 1.45 | 2.547 | 24.2 | 26.7 | 25.5 | 60.9 | 70.2 | 46.1 | 53.7 | 66.1 | 31.6 | 35.4 | 39.6 | 50.4 |
| TNL-L | 1.45 | 2.553 | 24.0 | 31.8 | 27.9 | 61.6 | 69.8 | 46.1 | 53.7 | 66.0 | 31.3 | 36.2 | 39.9 | 50.6 |
| *Tpe* | | | | | | | | | | | | | | |
| Mamba2 | 1.45 | 2.531 | 23.4 | 28.9 | 26.2 | 61.7 | 70.8 | 47.0 | 54.1 | 67.0 | 32.8 | 37.0 | 39.2 | 51.2 |
| GLA | 1.45 | 2.569 | 25.1 | 36.0 | 30.5 | 61.8 | 68.8 | 45.5 | 53.2 | 65.6 | 31.2 | 36.4 | 39.5 | 50.2 |
| Hgrn2 | 1.45 | 2.554 | 24.3 | 31.0 | 27.7 | 61.7 | 69.5 | 46.3 | 52.6 | 65.5 | 31.7 | 34.8 | 39.8 | 50.2 |
| LightNet | 1.45 | 2.567 | 24.4 | 31.1 | 27.8 | 61.1 | 69.4 | 45.3 | 52.8 | 64.9 | 33.1 | 35.8 | 40.1 | 50.3 |
| TNL | 1.45 | 2.556 | 24.3 | 29.6 | 27.0 | 61.1 | 70.5 | 46.2 | 52.3 | 65.9 | 31.1 | 35.4 | 40.3 | 50.4 |
| TNL-L | 1.45 | 2.550 | 24.0 | 30.8 | 27.4 | 61.7 | 69.9 | 45.9 | 51.9 | 67.3 | 31.6 | 35.8 | 40.3 | 50.6 |
| *Baseline* | | | | | | | | | | | | | | |
| LLaMA | 1.44 | 2.520 | 22.3 | 25.1 | 23.7 | 61.7 | 69.4 | 46.9 | 53.2 | 65.8 | 30.9 | 35.4 | 39.8 | 50.4 |

To elucidate the factors contributing to Mamba2's superior performance, we conducted an ablation study by decomposing its decay mechanism into the following variants: Mamba2 without **A** (denoted as Mamba2 w/o **A**), Mamba2 without Δ (Mamba2 w/o Δ), and Mamba2 without both **A** and Δ (Mamba2 w/o **A** & Δ)[1]. The results of this analysis are detailed in Table 2. Notably, Mamba2 w/o **A** demonstrated comparable or marginally improved performance relative to the original Mamba2. In contrast, Mamba2 w/o **A** & Δ exhibited a slight degradation in performance, while Mamba2 w/o Δ showed a significant decline.

To further investigate the underlying mechanisms, we evaluated the trained models on sequences of length 2048 from the "tinyshakespeare" dataset (Karpathy, 2023). We recorded the decay values from each network layer and analyzed their median distributions. As depicted in Figures 1, 4, and 6, our observations revealed the following insights:

---

[1]The taxonomy of decay mechanisms is comprehensively defined in Table 1.

- LightNet's median decay values are nearly 1, akin to linear attention without decay, causing attention dilution and thus lower performance.

- Mamba2's median decay values cluster around 0.8, consistently above 0.6, while Hgrn2 and GLA have layers with values near 0.2.

- Compared to Hgrn2, GLA exhibits significantly smaller decay values in later layers.

- Mamba2 w/o **A** has decay median values consistently above those of the original Mamba2;

- Mamba2 w/o $\Delta$ has very small decay median values, almost all below 0.4.

Combining the decay distribution analysis with model performance, we derive the following conclusions:

> **Takeaways for parameterization strategy**
>
> - Mamba2's decay mechanism performs best, and removing the parameter **A** does not degrade performance in most cases.
> - Decay values should be neither too small (close to 0) nor too large (close to 1), with median values around 0.8 providing optimal performance.

## 5.2 Parameter Sharing

We tested parameter sharing strategies ($\mathbf{k}_t = 1 - \lambda_t$) for Mamba2, GLA, Hgrn2, and LightNet. The results can be found in Table 2, Table 6, and Table 7. We found that parameter sharing has negligible impact on the performance of Mamba2 and Hgrn2, but significantly reduces the performance of GLA and LightNet. For further analysis, we visualized the median of decay values across layers in Figure 2, 5, and 7. We observed that the decay median values for Mamba2 and Hgrn2 mostly increased, with the number of layers having a median above 0.8 rising, while for GLA, the number of layers with a median above 0.8 decreased, which we suspect is the reason for GLA's degraded performance with parameter sharing. For LightNet, we calculated the overall average decay value and found that with parameter sharing, LightNet's average decay value increased from 0.97 to 0.99, making it closer to having no decay at all, thus resulting in worse performance. Combining the decay distribution with model performance, we conclude:

> **Takeaways for parameter sharing**
>
> - Parameter sharing cannot be used arbitrarily, as it may cause decay values to become too large or too small, thereby affecting performance.

## 5.3 Decay Granularity

We conducted a comparative analysis of Scalar Decay and Vector Decay across the models Mamba2, GLA, Hgrn2, and LightNet. Additionally, we extended our experimental framework to include TNL and TNL-L, the latter of which employs TNL's initialization but allows the parameters to be learnable. The results of these comparisons are detailed in Tables 2, 6 and 7. Our analysis revealed that, within the same parameterization strategy, Vector Decay consistently outperforms Scalar Decay. However, when different parameterization strategies are employed, Scalar Decay can, in certain cases, surpass the performance of Vector Decay.

To elucidate the underlying factors contributing to these performance differences, we investigated the relationship between the loss function and the median of all decay values within each model. Our findings indicated that:

- Scalar decay variants with better performance (compared to vector decay) typically have higher median values. For example, Mamba2 scalar decay has a higher median than Hgrn2 vector decay;

- TNL-L outperformed TNL, and surprisingly, the data-independent TNL and TNL-L were only slightly worse than Mamba2 but comparable to or better than data-dependent variants GLA and Hgrn2;

To analyze this, we visualized the decay medians and found that TNL's decay values are very close to 1 (but strictly less than 1), much larger than GLA and Hgrn2. TNL-L's decay median values are smaller than TNL, generally around 0.8.

Combining the decay distribution with model performance, we conclude:

> **Takeaways for decay granularity**
>
> - Under the same parameterization strategy, vector decay consistently outperforms scalar decay.
> - With different parameterization strategies, scalar decay can surpass vector decay, and the surpassing versions often exhibit larger median decay.
> - The range of decay values is more important than whether they are data-dependent, with the best-performing methods having decay medians around 0.8.

### 5.4 Compatibility with RPE

We investigated the compatibility of Scalar Decay with RoPE and TPE, as detailed in Tables 2, 6, and 7. Our findings indicate that, with the exception of LightNet, RoPE/TPE exhibited negligible impact on the models. This observation can be attributed to the fact that the majority of the methods, excluding LightNet, employ decay values less than 1. These sub-unity decay values inherently provide a locality prior, which substantially diminishes the influence of RoPE/TPE. In contrast, LightNet, characterized by decay values approaching 1, experiences attention dilution and consequently struggles to effectively perceive positional information. Based on these observations, we conclude:

> **Takeaways for compatibility with RPE**
>
> - For Linear Attention with decay values mostly less than 1, the effect of RoPE/TPE is negligible.

### 5.5 Proposed Simple Decay Parameterization

Based on the previous analysis, we propose a simple decay parameterization scheme (abbreviated as Simple Decay):

$$\lambda_t^j = \text{sigmoid}(\mathbf{f}_t^j + \Delta_t^j), \Delta_t^j \text{ initialize with argsigmoid}(p). \tag{4}$$

where parameter $p$ specifically represents the median decay value when the network is in its initialization state (assuming the median of $\mathbf{f}_t^j$ is 0). Note that this parameterization scheme is similar to Mamba2 without $\mathbf{A}$, with the difference being in the choice of $\Delta$, making this scheme more concise. Experimental results are shown in Table 3, 8. We conducted experiments in the vector decay scenario and observed that when $p = 0.95, 0.99$, the performance exceeds Mamba2, while at $p = 0.8, 0.9$, it underperforms compared to Mamba2. We visualize the distribution of decay values in Figure 8, 9. As can be observed, the median decay values after training increase as the initialization values increase. Additionally, the median decay values for most layers are smaller than the initial value $p$, indicating that the model tends to anneal from a high decay value to a more appropriate value.

> **Takeaways for compatibility with Simple Decay**
>
> - Simple decay with larger $p$ and Mamba2 Decay have comparable effects, with $p = 0.99$ achieving the best performance..

Table 3: Performance comparison of Mamba2 (M2) and Simple Decay (SD) with different initializations $p$. AVG represents average perplexity (lower is better) or average correct score rate.

| Me | p | Pa | Loss | PPL ↓ | | | Accuracy ↑ | | | | | | | | |
|---|---|---|---|---|---|---|---|---|---|---|---|---|---|---|---|
| | | | | Wiki | LMB | AVG | BOQA | PIQA | Hella | Wino | ARC-e | ARC-c | OBQA | SOQA | AVG |
| *1.45B models* | | | | | | | | | | | | | | | |
| M2 | - | 1.45 | 2.514 | 22.8 | 25.2 | 24.0 | 61.7 | 70.0 | 47.7 | 52.8 | 67.1 | 30.7 | 37.6 | 39.8 | 50.9 |
| SD | 0.8 | 1.45 | 2.516 | 22.9 | 24.5 | 23.7 | 61.7 | 70.4 | 47.8 | 53.0 | 66.1 | 32.3 | 36.8 | 39.8 | 51.0 |
| SD | 0.9 | 1.45 | 2.512 | 22.7 | 25.6 | 24.2 | 60.7 | 70.6 | 47.5 | 53.7 | 65.6 | 31.6 | 36.2 | 40.5 | 50.8 |
| SD | 0.95 | 1.45 | 2.511 | 22.7 | 23.9 | **23.3** | 62.1 | 70.2 | 48.1 | 51.1 | 66.0 | 32.4 | 35.4 | 41.3 | 50.8 |
| SD | 0.99 | 1.45 | 2.511 | 22.6 | 24.3 | 23.5 | 58.4 | 70.1 | 47.7 | 55.9 | 66.7 | 33.4 | 36.4 | 40.2 | **51.1** |

## 5.6 Extend to DPLR scenarios

In the previous experiments, we primarily focused on scenarios where the state transition matrix is diagonal form. In this section, we conduct experiments under the DPLR (Diagonal Plus Low-Rank) form, examining. As demonstrated in Table 4, 9, DPLR with no decay exhibits inferior performance across all metrics, including loss, average perplexity, and average accuracy. Under identical parameterization schemes, vector decay demonstrates superior efficacy compared to scalar decay. Increasing the parameter $p$ from 0 to 0.99 consistently yields lower loss and perplexity across all configurations, while zero-shot accuracy exhibits some variability. We hypothesize that this fluctuation may be attributed to the limited number of training tokens.

Based on these empirical observations, we draw the following conclusions:

---

**Takeaways for compatibility with DPLR**

- For DPLR models, Vector Decay achieves optimal performance, followed by scalar decay, with no decay yielding the poorest results.

- Simple Decay remains effective for DPLR model decay mechanisms, with larger $p$ values consistently producing lower loss.

---

Table 4: Performance comparison of different model methods under various configurations. AVG represents average perplexity (lower is better) or average correct score rate. No-D: DPLR with no decay, Sc-D-$p$: DPLR with scalar decay (simple decay style with value $p$), Ve-D: DPLR with vector decay (simple decay style with value $p$).

| Method | Pa | Loss | PPL ↓ | | | Accuracy ↑ | | | | | | | | |
|---|---|---|---|---|---|---|---|---|---|---|---|---|---|---|
| | | | Wiki | LMB | AVG | BOQA | PIQA | Hella | Wino | ARC-e | ARC-c | OBQA | SOQA | AVG |
| No-D | 1.45 | 2.591 | 23.7 | 31.1 | 27.4 | 61.3 | 69.3 | 44.3 | 53.0 | 65.3 | 31.2 | 34.8 | 39.6 | 49.9 |
| Sc-D-0 | 1.45 | 2.523 | 23.1 | 26.6 | 24.8 | 61.5 | 70.0 | 47.1 | 53.1 | 65.4 | 33.1 | 35.4 | 40.8 | 50.8 |
| Sc-D-0.99 | 1.45 | 2.507 | 22.4 | 23.1 | 22.8 | 61.4 | 71.0 | 47.4 | 53.8 | 65.5 | 31.9 | 36.8 | 40.0 | 51.0 |
| Ve-D-0 | 1.47 | 2.508 | 22.5 | 22.3 | 22.4 | 60.8 | 69.6 | 48.1 | 53.5 | 66.9 | 32.7 | 36.0 | 40.0 | 51.0 |
| Ve-D-0.99 | 1.47 | 2.498 | 22.0 | 21.2 | 21.6 | 60.9 | 69.8 | 48.4 | 54.3 | 66.5 | 32.6 | 34.2 | 40.5 | 50.9 |

## 6 Conclusion

In this paper, we presented a comprehensive analysis of decay mechanisms in Linear Attention, exploring a design space with four key dimensions: parameterization strategy, parameter sharing, decay granularity, and compatibility with positional encoding. Through standardized experiments, we found that decay mechanisms significantly affect model performance, yielding several insights. Building upon these findings, we propose Simple Decay, a streamlined parameterization scheme that balances strong performance with reduced complexity. Our study underscores the critical role of well-configured decay in sequence modeling and provides practical guidance for designing efficient Linear Attention mechanisms. Future research could investigate the applicability of these insights to larger models and diverse downstream tasks.

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

# A Appendix

## A.1 Model Architecture

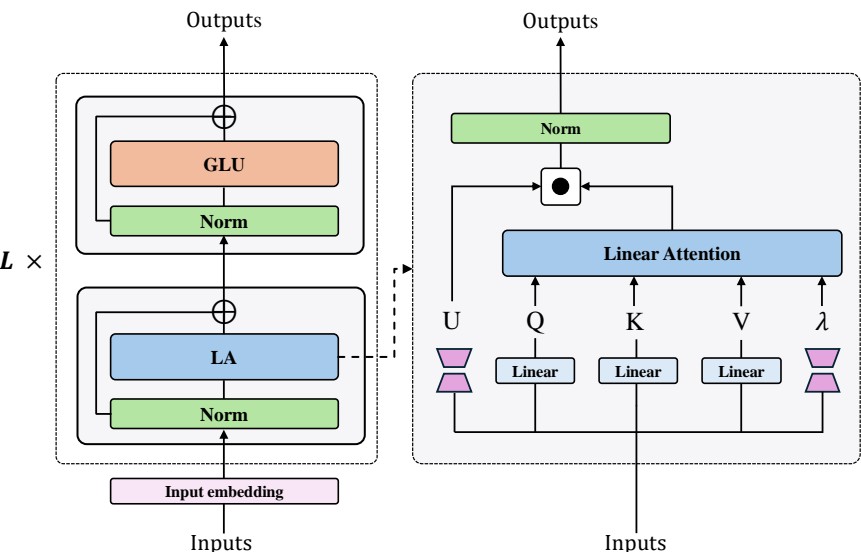

Figure 3: **Model Architecture.** Model architecture diagram of Decay Linear Transformer: Each Decay Linear Transformer consists of multiple Decay Linear Transformer Layers, with each Layer comprising Decay Linear Attention and GLU; for Decay Linear Attention, its computational logic is shown in the right figure.

## A.2 Decay computation

For decay, we first obtain activation $\mathbf{F}^j$ through linear layers, then calculate $\lambda_t^j$ through function $f$ (whose form is determined by the **Parameterization Strategy**, see Table 1). For vector decay, we use low-rank mapping to minimize the impact of parameter count on our conclusions (when comparing scalar decay and vector decay without low-rank mapping, the former would have $d^2 - dh$ fewer parameters than the latter, which is far greater than the difference when using low-rank mapping: $2d(d/h) - dh$). The detailed computation is listed as E.q. 5:

$$
\mathbf{F}^j = \begin{cases} \mathbf{X}\mathbf{W}_{d_1}^j, & \text{scalar decay,} \\ \mathbf{X}\mathbf{W}_{d_2}^j\mathbf{W}_{d_3}^j, & \text{vector decay with out parameter sharing,} \\ \mathbf{X}\mathbf{W}_{d_4}^j, & \text{vector decay with parameter sharing,} \end{cases} \tag{5}
$$

$$
\lambda_t^j = f(\mathbf{f}_t^j), \mathbf{W}_{d_1}^j \in \mathbb{R}^{d \times 1}, \mathbf{W}_{d_2}^j \in \mathbb{R}^{d \times d/h}, \mathbf{W}_{d_3}^j \in \mathbb{R}^{d/h \times d/h}, \mathbf{W}_{d_4}^j \in \mathbb{R}^{d \times d/h}, j = 1, \dots, h.
$$

## A.3 Compatibility between Decay and RoPE

In subsequent discussions, we omit the head superscript $j$ to simplify notation.

Assuming we apply RoPE to $\mathbf{q}_t, \mathbf{k}_t$ to obtain $\bar{\mathbf{q}}_t, \bar{\mathbf{k}}_t$:

$$
\bar{\mathbf{y}}_t = \mathbf{R}_t\mathbf{y}_t \in \mathbb{R}^d, \mathbf{R}_t = \text{diag}(\mathbf{R}_{t,1}, \dots, \mathbf{R}_{t,d/2}), \mathbf{R}_{t,j} = \begin{bmatrix} \cos(t\theta_j) & -\sin(t\theta_j) \\ \sin(t\theta_j) & \cos(t\theta_j) \end{bmatrix}, \mathbf{y} \in \{\mathbf{q}, \mathbf{k}\}. \tag{6}
$$

Then according to the Linear Attention recurrence equation:

$$\mathbf{s}_t = \mathrm{diag}(\lambda_t)\mathbf{s}_{t-1} + \bar{\mathbf{k}}_t\mathbf{v}_t^\top,$$

$$\gamma_t \triangleq \prod_{j=1}^{t}\lambda_j,$$

$$\mathbf{s}_t = \mathrm{diag}(\gamma_t)\sum_{j=1}^{t}\mathrm{diag}(1/\gamma_j)\bar{\mathbf{k}}_j\mathbf{v}_j^\top,$$

$$\mathbf{o}_t^\top = \bar{\mathbf{q}}_t^\top\mathbf{s}_t \tag{7}$$

$$= \bar{\mathbf{q}}_t^\top\mathrm{diag}(\gamma_t)\sum_{j=1}^{t}\mathrm{diag}(1/\gamma_j)\bar{\mathbf{k}}_j\mathbf{v}_j^\top$$

$$= (\mathrm{diag}(\gamma_t)\mathbf{R}_t\mathbf{q}_t)^\top\sum_{j=1}^{t}\mathrm{diag}(1/\gamma_j)\mathbf{R}_j\mathbf{k}_j\mathbf{v}_j^\top.$$

When the elements of $\gamma_t$ are all identical (scalar decay), the above expression can be simplified to:

$$\mathbf{o}_t^\top = (\mathrm{diag}(\gamma_t)\mathbf{R}_t\mathbf{q}_t)^\top\sum_{j=1}^{t}\mathrm{diag}(1/\gamma_j)\mathbf{R}_j\mathbf{k}_j\mathbf{v}_j^\top$$

$$= \gamma_t(\mathbf{R}_t\mathbf{q}_t)^\top\sum_{j=1}^{t}(1/\gamma_j)\mathbf{R}_j\mathbf{k}_j\mathbf{v}_j^\top$$

$$= \mathbf{q}_t^\top\sum_{j=1}^{t}(\gamma_t/\gamma_j)\mathbf{R}_t^\top\mathbf{R}_j\mathbf{k}_j\mathbf{v}_j^\top \tag{8}$$

$$= \mathbf{q}_t^\top\sum_{j=1}^{t}(\gamma_t/\gamma_j)\mathbf{R}_{t-j}\mathbf{k}_j\mathbf{v}_j^\top.$$

When $\gamma_t$ is vector decay, assuming $\gamma_t$ has the form:

$$\gamma_t^\top = [\gamma_{t,1}, \gamma_{t,1}, \ldots, \gamma_{t,d/2}, \gamma_{t,d/2}] \in \mathbb{R}^d. \tag{9}$$

Since $\mathbf{R}_t$ is a block-diagonal matrix with block size 2, for each block, $\gamma_t$ acts as scalar decay, so it also satisfies:

$$\mathbf{o}_t^\top = \mathbf{q}_t^\top\sum_{j=1}^{t}(\gamma_t/\gamma_j)\mathbf{R}_{t-j}\mathbf{k}_j\mathbf{v}_j^\top. \tag{10}$$

Since vector decay requires special design to satisfy RoPE's relative positional properties (reducing decay's degrees of freedom by half), to simplify the problem, all our experiments are conducted with scalar decay.

### A.4 Configuration

Table 5: Model configurations for different parameter sizes.

| Params(B) | Layers | Hidden Dim | Num Heads | L.R. | Batch Size | SeqLen | GPUs |
|-----------|--------|------------|-----------|-------|------------|--------|------|
| 0.16 | 12 | 768 | 12 | 3E-04 | 32 | 2048 | 8 |
| 0.41 | 24 | 1024 | 16 | 3E-04 | 32 | 2048 | 8 |
| 1.45 | 24 | 2048 | 32 | 3E-04 | 32 | 2048 | 8 |

### A.5 More experimental results

### A.6 More visualization results

Table 6: Performance comparison of different model methods under various configurations. AVG represents average perplexity (lower is better) or average correct score rate.

| Method | Pa | Loss | PPL ↓ | | | Accuracy ↑ | | | | | | | | |
|---|---|---|---|---|---|---|---|---|---|---|---|---|---|---|
| | | | Wiki | LMB | AVG | BOQA | PIQA | Hella | Wino | ARC-e | ARC-c | OBQA | SOQA | AVG |
| *Vector decay* | | | | | | | | | | | | | | |
| Mamba2 | 0.16 | 2.947 | 40.1 | 92.9 | 66.5 | 60.4 | 63.6 | 33.3 | 51.4 | 54.5 | 24.9 | 31.2 | 38.6 | 44.7 |
| GLA | 0.16 | 2.975 | 42.2 | 131.3 | 86.7 | 60.0 | 63.6 | 33.3 | 48.9 | 53.1 | 26.8 | 31.0 | 37.0 | 44.2 |
| Hgrn2 | 0.16 | 2.966 | 41.5 | 107.2 | 74.4 | 60.6 | 64.3 | 33.0 | 50.3 | 52.5 | 24.6 | 29.6 | 37.5 | 44.0 |
| LightNet | 0.16 | 3.027 | 51.7 | 173.3 | 112.5 | 61.1 | 62.1 | 30.4 | 50.4 | 51.7 | 24.5 | 30.2 | 35.3 | 43.2 |
| *Mamba ablation* | | | | | | | | | | | | | | |
| Mamba2 w/o **A** | 0.16 | 2.946 | 39.7 | 98.8 | 69.2 | 58.0 | 64.1 | 33.3 | 51.6 | 53.8 | 25.7 | 30.8 | 37.2 | 44.3 |
| Mamba2 w/o Δ | 0.16 | 3.019 | 44.9 | 132.1 | 88.5 | 61.6 | 62.8 | 31.8 | 49.0 | 53.4 | 24.4 | 31.0 | 36.7 | 43.8 |
| Mamba2 w/o **A**,Δ | 0.16 | 2.973 | 42.3 | 116.4 | 79.3 | 59.6 | 63.7 | 32.4 | 49.8 | 52.6 | 25.5 | 31.2 | 37.0 | 44.0 |
| *Parameter share* | | | | | | | | | | | | | | |
| Mamba2 | 0.16 | 2.947 | 39.9 | 91.5 | 65.7 | 60.7 | 63.9 | 33.3 | 49.3 | 54.6 | 27.3 | 31.4 | 38.6 | 44.9 |
| GLA | 0.16 | 3.048 | 46.9 | 180.7 | 113.8 | 61.0 | 64.0 | 32.2 | 49.3 | 53.2 | 25.0 | 30.8 | 36.3 | 44.0 |
| Hgrn2 | 0.16 | 2.966 | 40.9 | 94.2 | 67.6 | 58.1 | 63.6 | 33.2 | 51.9 | 54.2 | 25.9 | 31.6 | 37.1 | 44.4 |
| LightNet | 0.16 | 3.104 | 48.1 | 312.6 | 180.3 | 61.1 | 62.5 | 30.5 | 50.2 | 49.7 | 24.7 | 30.2 | 35.7 | 43.1 |
| *Scalar decay* | | | | | | | | | | | | | | |
| Mamba2 | 0.16 | 2.960 | 41.0 | 104.7 | 72.9 | 58.6 | 63.2 | 33.1 | 50.1 | 54.1 | 25.9 | 30.2 | 38.0 | 44.2 |
| GLA | 0.16 | 3.008 | 43.8 | 128.0 | 85.9 | 57.2 | 63.7 | 32.4 | 51.4 | 53.8 | 25.7 | 30.4 | 37.7 | 44.0 |
| Hgrn2 | 0.16 | 2.987 | 44.6 | 181.5 | 113.0 | 59.1 | 63.2 | 32.4 | 51.4 | 52.6 | 25.1 | 30.2 | 37.7 | 44.0 |
| LightNet | 0.16 | 3.032 | 44.6 | 149.4 | 97.0 | 59.0 | 62.7 | 31.4 | 52.6 | 53.2 | 25.3 | 30.2 | 36.0 | 43.8 |
| TNL | 0.16 | 2.985 | 42.4 | 117.5 | 79.9 | 61.9 | 63.2 | 32.4 | 49.3 | 54.8 | 26.9 | 31.2 | 37.8 | 44.7 |
| TNL-L | 0.16 | 2.970 | 41.3 | 118.7 | 80.0 | 61.2 | 64.0 | 32.6 | 50.8 | 54.5 | 23.5 | 33.6 | 37.3 | 44.7 |
| *Rope* | | | | | | | | | | | | | | |
| Mamba2 | 0.16 | 2.959 | 40.9 | 110.7 | 75.8 | 59.4 | 63.9 | 32.9 | 50.4 | 54.3 | 26.3 | 29.8 | 37.2 | 44.3 |
| GLA | 0.16 | 3.010 | 44.9 | 185.3 | 115.1 | 57.7 | 63.7 | 32.6 | 51.1 | 52.9 | 24.2 | 32.4 | 37.6 | 44.0 |
| Hgrn2 | 0.16 | 2.990 | 43.1 | 128.3 | 85.7 | 56.9 | 63.1 | 32.6 | 50.5 | 51.8 | 25.1 | 32.2 | 36.4 | 43.6 |
| LightNet | 0.16 | 3.002 | 42.5 | 128.7 | 85.6 | 60.9 | 63.4 | 31.8 | 50.1 | 53.5 | 25.9 | 29.6 | 36.8 | 44.0 |
| TNL | 0.16 | 2.975 | 41.7 | 109.0 | 75.3 | 62.0 | 63.1 | 32.3 | 49.5 | 53.9 | 24.6 | 28.6 | 37.1 | 43.9 |
| TNL-L | 0.16 | 2.972 | 41.4 | 111.0 | 76.2 | 57.0 | 64.5 | 32.5 | 51.5 | 53.9 | 25.3 | 30.2 | 37.7 | 44.1 |
| *Tpe* | | | | | | | | | | | | | | |
| Mamba2 | 0.16 | 2.931 | 38.8 | 95.6 | 67.2 | 53.5 | 63.9 | 33.8 | 51.5 | 53.5 | 25.6 | 32.0 | 36.5 | 43.8 |
| GLA | 0.16 | 2.986 | 43.0 | 155.3 | 99.2 | 61.2 | 63.8 | 32.2 | 50.7 | 53.1 | 26.8 | 33.2 | 37.2 | 44.8 |
| Hgrn2 | 0.16 | 2.969 | 41.5 | 98.8 | 70.1 | 60.7 | 63.9 | 33.0 | 52.4 | 53.4 | 25.9 | 31.2 | 37.2 | 44.7 |
| LightNet | 0.16 | 2.988 | 41.6 | 114.6 | 78.1 | 56.1 | 63.8 | 32.1 | 50.6 | 53.2 | 25.9 | 30.8 | 37.1 | 43.7 |
| TNL-L | 0.16 | 2.948 | 40.0 | 108.0 | 74.0 | 60.0 | 63.9 | 33.3 | 51.5 | 54.3 | 26.6 | 31.0 | 37.6 | 44.8 |
| *Baseline* | | | | | | | | | | | | | | |
| LLaMA | 0.16 | 2.921 | 37.0 | 87.5 | 62.2 | 60.6 | 64.1 | 32.8 | 48.5 | 53.7 | 25.8 | 30.6 | 36.7 | 44.1 |

Table 7: Performance comparison of different model methods under various configurations. AVG represents average perplexity (lower is better) or average correct score rate.

| Method | Pa | Loss | PPL ↓ | | | Accuracy ↑ | | | | | | | | |
|---|---|---|---|---|---|---|---|---|---|---|---|---|---|---|
| | | | Wiki | LMB | AVG | BOQA | PIQA | Hella | Wino | ARC-e | ARC-c | OBQA | SOQA | AVG |
| *Vector decay* | | | | | | | | | | | | | | |
| Mamba2 | 0.42 | 2.720 | 29.8 | 46.8 | 38.3 | 61.2 | 67.1 | 39.5 | 49.6 | 60.1 | 28.3 | 32.2 | 38.6 | 47.1 |
| GLA | 0.42 | 2.743 | 31.0 | 56.5 | 43.8 | 58.5 | 67.4 | 39.5 | 50.9 | 60.0 | 27.3 | 34.6 | 38.4 | 47.1 |
| Hgrn2 | 0.42 | 2.736 | 30.4 | 45.3 | 37.9 | 58.4 | 66.3 | 39.2 | 50.8 | 58.9 | 28.1 | 33.0 | 39.1 | 46.7 |
| LightNet | 0.42 | 2.784 | 31.2 | 55.6 | 43.4 | 60.8 | 66.9 | 38.0 | 50.8 | 58.9 | 27.0 | 31.2 | 38.7 | 46.5 |
| *Mamba ablation* | | | | | | | | | | | | | | |
| Mamba2 w/o $\mathbf{A}$ | 0.42 | 2.717 | 29.5 | 45.4 | 37.5 | 61.1 | 66.5 | 39.7 | 53.9 | 61.5 | 28.8 | 33.2 | 39.7 | 48.1 |
| Mamba2 w/o $\Delta$ | 0.42 | 2.793 | 33.6 | 65.4 | 49.5 | 61.5 | 66.3 | 37.4 | 49.5 | 57.8 | 27.0 | 33.0 | 37.2 | 46.2 |
| Mamba2 w/o $\mathbf{A},\Delta$ | 0.42 | 2.738 | 30.7 | 51.8 | 41.3 | 57.8 | 66.5 | 39.4 | 51.9 | 58.5 | 27.4 | 34.8 | 37.7 | 46.7 |
| *Parameter share* | | | | | | | | | | | | | | |
| Mamba2 | 0.41 | 2.719 | 29.5 | 45.7 | 37.6 | 61.7 | 67.0 | 39.6 | 50.4 | 60.9 | 28.9 | 33.4 | 38.4 | 47.5 |
| GLA | 0.41 | 2.805 | 34.3 | 79.0 | 56.7 | 59.1 | 65.4 | 37.6 | 52.7 | 60.4 | 28.0 | 32.4 | 39.1 | 46.8 |
| Hgrn2 | 0.41 | 2.738 | 30.5 | 45.1 | 37.8 | 61.5 | 66.8 | 38.9 | 51.9 | 60.7 | 28.1 | 34.2 | 38.5 | 47.6 |
| LightNet | 0.41 | 2.859 | 34.5 | 126.8 | 80.6 | 61.0 | 63.9 | 35.2 | 52.0 | 55.9 | 25.4 | 32.8 | 37.7 | 45.5 |
| *Scalar decay* | | | | | | | | | | | | | | |
| Mamba2 | 0.42 | 2.733 | 30.3 | 50.8 | 40.6 | 61.9 | 67.0 | 39.4 | 50.2 | 59.1 | 28.0 | 32.4 | 37.8 | 47.0 |
| GLA | 0.42 | 2.769 | 31.4 | 56.4 | 43.9 | 56.7 | 67.3 | 38.5 | 53.0 | 59.0 | 27.8 | 34.2 | 38.5 | 46.9 |
| Hgrn2 | 0.42 | 2.753 | 32.3 | 70.1 | 51.2 | 60.7 | 67.0 | 38.5 | 51.7 | 58.7 | 27.0 | 32.0 | 38.7 | 46.8 |
| LightNet | 0.42 | 2.787 | 32.2 | 61.0 | 46.6 | 61.4 | 65.9 | 37.0 | 50.4 | 59.2 | 26.6 | 31.0 | 38.3 | 46.2 |
| TNL | 0.41 | 2.759 | 31.9 | 50.6 | 41.3 | 58.9 | 66.2 | 38.3 | 51.6 | 60.3 | 28.0 | 33.6 | 37.8 | 46.8 |
| TNL-L | 0.41 | 2.747 | 30.8 | 54.9 | 42.8 | 61.6 | 67.7 | 38.4 | 52.2 | 59.6 | 27.7 | 32.2 | 38.7 | 47.3 |
| *Rope* | | | | | | | | | | | | | | |
| Mamba2 | 0.42 | 2.732 | 30.4 | 50.0 | 40.2 | 61.4 | 67.1 | 39.4 | 50.2 | 60.0 | 28.2 | 34.2 | 39.0 | 47.4 |
| GLA | 0.42 | 2.764 | 32.4 | 67.5 | 50.0 | 60.7 | 66.2 | 38.5 | 49.6 | 59.4 | 27.8 | 31.8 | 38.5 | 46.6 |
| Hgrn2 | 0.42 | 2.751 | 31.6 | 53.4 | 42.5 | 57.9 | 66.8 | 38.6 | 50.5 | 59.5 | 28.8 | 33.8 | 39.0 | 46.9 |
| LightNet | 0.42 | 2.757 | 31.3 | 53.8 | 42.6 | 61.0 | 66.3 | 38.5 | 51.2 | 59.3 | 27.8 | 33.6 | 38.5 | 47.0 |
| TNL | 0.41 | 2.749 | 31.4 | 53.7 | 42.6 | 61.5 | 66.4 | 38.8 | 51.4 | 59.9 | 28.1 | 34.8 | 38.4 | 47.4 |
| TNL-L | 0.41 | 2.740 | 30.7 | 54.5 | 42.6 | 60.7 | 66.8 | 39.0 | 50.8 | 60.0 | 29.0 | 34.2 | 38.3 | 47.3 |
| *Tpe* | | | | | | | | | | | | | | |
| Mamba2 | 0.42 | 2.712 | 29.5 | 47.2 | 38.4 | 60.3 | 67.6 | 39.8 | 50.6 | 60.7 | 29.1 | 32.8 | 39.7 | 47.6 |
| GLA | 0.42 | 2.751 | 31.4 | 70.1 | 50.7 | 60.0 | 65.8 | 39.2 | 50.9 | 60.0 | 26.4 | 34.4 | 38.6 | 46.9 |
| Hgrn2 | 0.42 | 2.730 | 41.5 | 98.8 | 70.1 | 60.7 | 63.9 | 33.0 | 52.4 | 53.4 | 25.9 | 31.2 | 37.2 | 44.7 |
| LightNet | 0.42 | 2.754 | 31.1 | 59.4 | 45.2 | 61.6 | 67.1 | 38.6 | 49.4 | 59.6 | 28.1 | 31.2 | 39.1 | 46.8 |
| TNL | 0.42 | 2.740 | 30.9 | 50.7 | 40.8 | 61.0 | 67.7 | 39.5 | 51.2 | 61.5 | 29.1 | 33.0 | 39.4 | 47.8 |
| TNL-L | 0.42 | 2.725 | 30.0 | 47.3 | 38.6 | 60.8 | 67.0 | 39.4 | 49.6 | 61.4 | 29.0 | 33.6 | 39.1 | 47.5 |
| *Baseline* | | | | | | | | | | | | | | |
| LLaMA | 0.41 | 2.720 | 28.5 | 46.7 | 37.6 | 60.7 | 66.7 | 38.9 | 51.6 | 58.6 | 28.2 | 33.4 | 39.0 | 47.1 |

Table 8: Performance comparison of Mamba2(M2) and Simple Decay (SD) with different initializations $p$. AVG represents average perplexity (lower is better) or average correct score rate.

| Me | $p$ | Pa | Loss | PPL ↓ | | | Accuracy ↑ | | | | | | | | |
|---|---|---|---|---|---|---|---|---|---|---|---|---|---|---|---|
| | | | | Wiki | LMB | AVG | BOQA | PIQA | Hella | Wino | ARC-e | ARC-c | OBQA | SOQA | AVG |
| *160M models* | | | | | | | | | | | | | | | |
| M2 | - | 0.16 | 2.947 | 40.1 | 92.9 | 66.5 | 60.4 | 63.6 | 33.3 | 51.4 | 54.5 | 24.9 | 31.2 | 38.6 | 44.7 |
| SD | 0.8 | 0.16 | 2.954 | 41.0 | 117.6 | 79.3 | 61.0 | 63.6 | 33.4 | 50.1 | 54.8 | 25.6 | 30.8 | 36.5 | 44.5 |
| SD | 0.9 | 0.16 | 2.949 | 40.6 | 105.6 | 73.1 | 62.0 | 64.0 | 33.1 | 50.8 | 53.7 | 26.5 | 31.0 | 37.8 | 44.8 |
| SD | 0.95 | 0.16 | 2.939 | 39.7 | 97.5 | 68.6 | 59.5 | 64.2 | 33.5 | 49.2 | 54.4 | 26.4 | 31.6 | 37.1 | 44.5 |
| SD | 0.99 | 0.16 | 2.940 | 39.4 | 96.7 | 68.0 | 61.3 | 63.9 | 33.4 | 48.8 | 53.8 | 24.7 | 32.0 | 36.7 | 44.3 |
| *410M models* | | | | | | | | | | | | | | | |
| M2 | - | 0.42 | 2.720 | 29.8 | 46.8 | 38.3 | 61.2 | 67.1 | 39.5 | 49.6 | 60.1 | 28.3 | 32.2 | 38.6 | 47.1 |
| SD | 0.8 | 0.42 | 2.727 | 30.2 | 45.3 | 37.8 | 59.8 | 67.7 | 40.0 | 51.1 | 59.3 | 29.3 | 34.6 | 38.7 | 47.6 |
| SD | 0.9 | 0.42 | 2.722 | 29.8 | 45.6 | 37.7 | 61.0 | 68.1 | 40.1 | 51.2 | 59.3 | 27.2 | 30.6 | 39.1 | 47.1 |
| SD | 0.95 | 0.42 | 2.716 | 29.5 | 48.9 | 39.2 | 60.5 | 67.0 | 39.7 | 52.8 | 60.1 | 27.4 | 34.0 | 39.2 | 47.6 |
| SD | 0.99 | 0.42 | 2.711 | 29.4 | 46.7 | 38.1 | 61.0 | 66.8 | 40.0 | 50.7 | 60.9 | 28.9 | 33.6 | 38.7 | 47.6 |

Table 9: Performance comparison of different model methods under various configurations. AVG represents average perplexity (lower is better) or average correct score rate. No-D: DPLR with no decay, Sc-D: DPLR with scalar decay, Ve-D: DPLR with vector decay.

| Method | Pa | Loss | PPL ↓ | | | Accuracy ↑ | | | | | | | | |
|---|---|---|---|---|---|---|---|---|---|---|---|---|---|---|
| | | | Wiki | LMB | AVG | BOQA | PIQA | Hella | Wino | ARC-e | ARC-c | OBQA | SOQA | AVG |
| No-D | 0.16 | 3.000 | 41.1 | 120.0 | 80.6 | 56.4 | 62.7 | 31.7 | 49.5 | 52.9 | 26.6 | 31.0 | 36.7 | 43.4 |
| Sc-D-0 | 0.16 | 2.965 | 40.7 | 121.8 | 81.3 | 60.1 | 64.1 | 32.7 | 50.0 | 53.6 | 25.3 | 30.8 | 36.7 | 44.2 |
| Sc-D-0.99 | 0.16 | 2.941 | 39.0 | 103.7 | 71.4 | 60.1 | 64.0 | 33.0 | 50.9 | 53.8 | 25.0 | 31.0 | 38.1 | 44.5 |
| Ve-D-0 | 0.17 | 2.937 | 39.0 | 83.4 | 61.2 | 61.0 | 63.9 | 33.7 | 50.6 | 54.9 | 25.2 | 31.4 | 38.6 | 44.9 |
| Ve-D-0.99 | 0.17 | 2.920 | 37.9 | 73.5 | 55.7 | 60.1 | 64.9 | 33.8 | 48.2 | 53.6 | 25.3 | 30.8 | 36.5 | 44.2 |
| No-D | 0.42 | 2.773 | 30.8 | 58.3 | 44.6 | 59.7 | 66.6 | 37.7 | 50.7 | 58.0 | 28.0 | 32.6 | 37.4 | 46.3 |
| Sc-D-0 | 0.42 | 2.736 | 30.1 | 56.1 | 43.1 | 58.4 | 67.6 | 39.4 | 51.9 | 58.4 | 27.1 | 33.6 | 36.8 | 46.7 |
| Sc-D-0.99 | 0.42 | 2.717 | 29.1 | 46.0 | 37.5 | 61.1 | 67.6 | 39.5 | 51.1 | 61.2 | 29.7 | 34.0 | 38.7 | 47.9 |
| Ve-D-0 | 0.42 | 2.732 | 29.3 | 45.2 | 37.3 | 61.0 | 67.4 | 39.7 | 50.4 | 59.6 | 29.5 | 32.6 | 37.7 | 47.2 |
| Ve-D-0.99 | 0.42 | 2.719 | 28.5 | 43.3 | 35.9 | 60.7 | 67.0 | 40.0 | 50.3 | 60.3 | 27.7 | 34.6 | 38.5 | 47.4 |

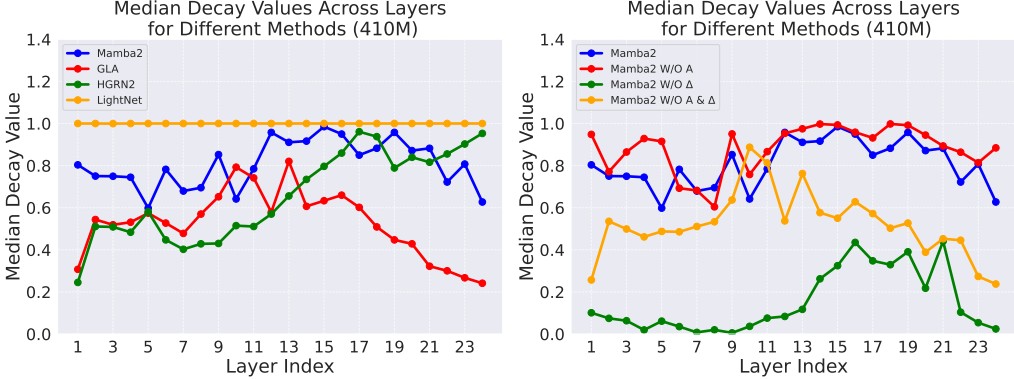

Figure 4: Distribution of median decay values for each layer across different methods, with model size of 410M. **Left figure**: Median distribution of Vector Decay. **Right figure**: Median distribution of Mamba ablation under Vector Decay.

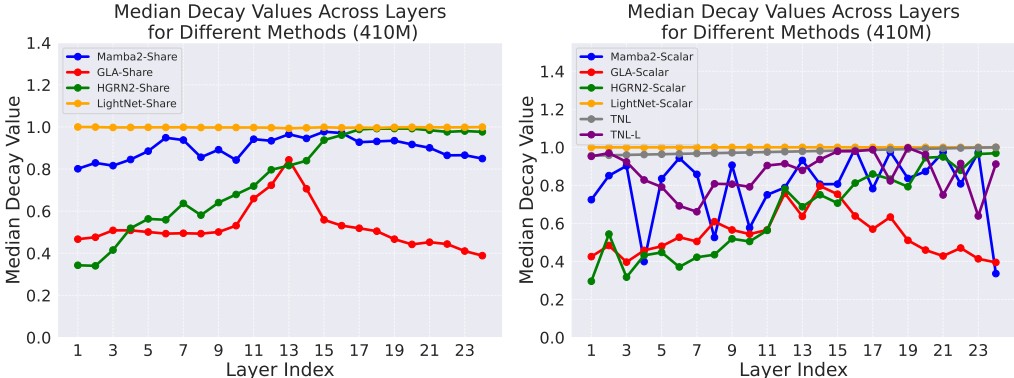

Figure 5: Distribution of median decay values for each layer across different methods, with model size of 410M. **Left figure**: Median distribution of Share Decay. **Right figure**: Median distribution of Scalar Decay.

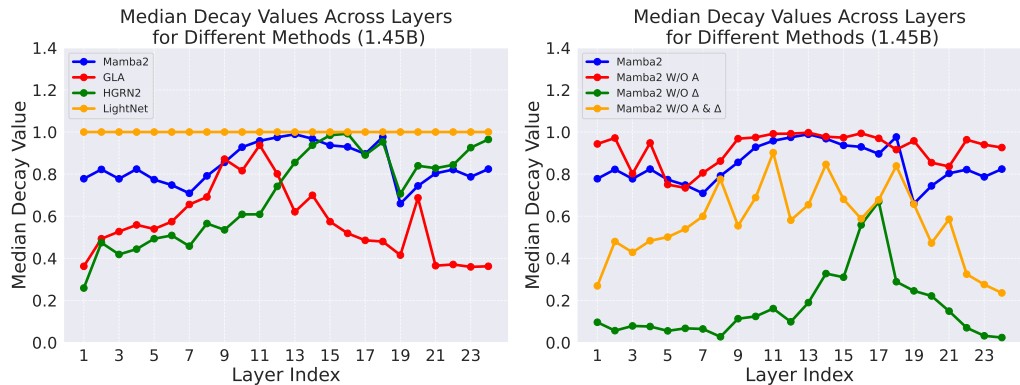

Figure 6: Distribution of median decay values for each layer across different methods, with model size of 1.45B. **Left figure**: Median distribution of Vector Decay. **Right figure**: Median distribution of Mamba ablation under Vector Decay.

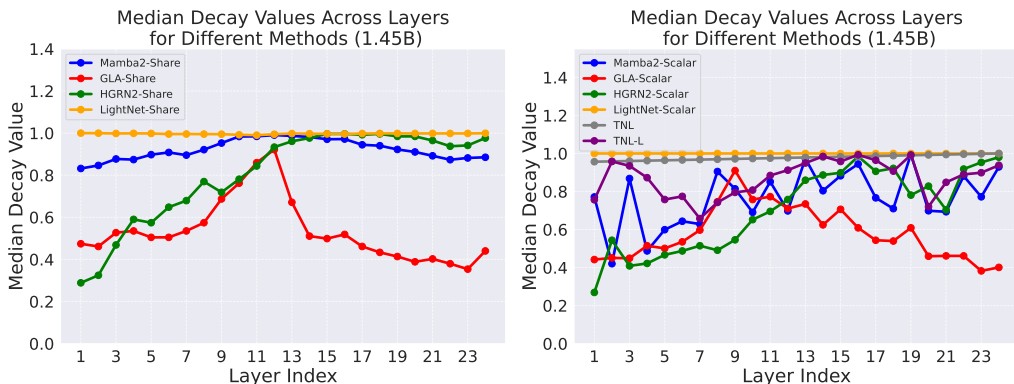

Figure 7: Distribution of median decay values for each layer across different methods, with model size of 1.45B. **Left figure**: Median distribution of Share Decay. **Right figure**: Median distribution of Scalar Decay.

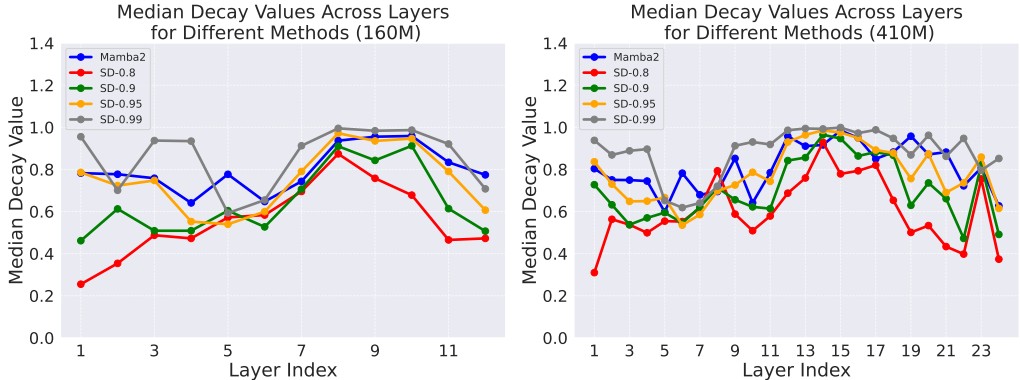

Figure 8: Visualization of median decay values for each layer in Simple Decay with different $p$ initializations, for model sizes of 160M and 410M.

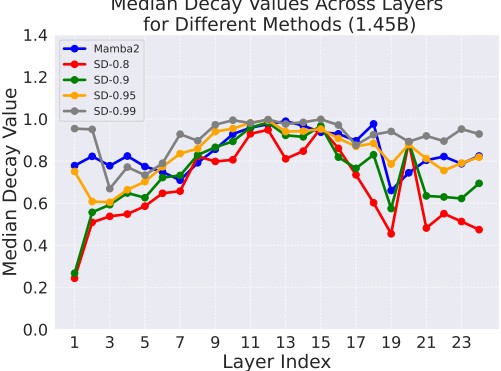

Figure 9: Visualization of median decay values for each layer in Simple Decay with different $p$ initializations, for model sizes of 1.45B.

