# OpenReview forum: "Elucidating the Design Space of Decay in Linear Attention"
_colmweb.org/COLM/2025/Conference — COLM 2025_

### Official Review · Reviewer_R6Me · 2025-04-14

**Rating:** 7
**Confidence:** 2
**Ethics Flag:** 1

**Summary:**

In this paper, the authors present an investigation of the decay mechanisms in linear attention models. Linear attention models are used as alternatives to the transformer architecture. As presented in the paper, the authors analyze four aspects of the design space of decay mechanisms:

$\bullet$ Parameterization strategy: How decay values are calculated (static, trainable, data-dependent).

$\bullet$ Parameter sharing: Whether to allocate dedicated parameters for decay computation.

$\bullet$ Decay granularity: Whether to use uniform scalar decay across dimensions or fine-grained vector decay.

$\bullet$ Positional encoding integration: How decay mechanisms interact with positional information, particularly RoPE.

**Questions To Authors:**

See "Reasons To Reject".

**Reasons To Accept:**

The reasons to accept this paper are summarized as follows:

$\bullet$ This paper gives a very comprehensive analysis of the decay mechanisms through parameterization strategy, parameter sharing, decay granularity, and positional encoding integration. It clearly shows the strengths and limitations.

$\bullet$ Experimental results shown in this paper are well-presented. They include different models, such as 60M, 410M, 1.45B parameters, performing on different tasks.

$\bullet$ This paper has novel contributions: the proposed Simple Decay mechanism demonstrates the practical application of the research findings.

**Reasons To Reject:**

The reasons to reject this paper are summarized as follows:

$\bullet$ While this paper gives comprehensive experimental results, its theoretical contributions are rather limited. It is better if the author can provide more theoretical analysis on why certain decay configurations perform better.

$\bullet$ The paper focuses on performance metrics but doesn't thoroughly analyze the computational cost differences between decay mechanisms.

---

> ### Author Response · Authors · 2025-05-31
> **Response to Reviewer R6Me**
>
> Q1. Theoretical analysis of different decay parameterizations
>
> R1. We provide visualizations of the decay median in Figures 1 and 2 of the paper. From a theoretical standpoint, the decay mechanism functions analogously to a soft sliding window, influencing the model’s effective receptive field. Specifically, a decay value close to 0 results in an overly narrow receptive field, which is suboptimal for sequence modeling. Conversely, a decay value close to 1 fails to enforce sufficient locality, thereby limiting the model’s ability to effectively capture language-specific patterns using linear attention mechanisms.
>
> Q2. Computational cost overhead of different decay parameterization schemes
>
> R2. we report the running speeds under different parameterization schemes using the TGS (tokens per GPU per second) metric. As shown, all methods—except for LightNet—exhibit comparable TGS values.
> | Method | Params | TGS |
> |----------|--------|---------|
> | Mamba2 | 1.4525 | 18330.1 |
> | GLA | 1.4524 | 18354.5 |
> | Hgrn2 | 1.4525 | 18386.1 |
> | LightNet | 1.4524 | 16992.2 |
> | SD | 1.4525 | 18381.2 |
>
> This discrepancy is attributable to the computational overhead introduced by LightNet, which requires the use of the `logcumsumexp` operation, as detailed in Table 1. In contrast, the other methods involve only element-wise operations, leading to more efficient computation and similar runtime performance.

---

### Official Review · Reviewer_wY2B · 2025-05-11

**Rating:** 6
**Confidence:** 3
**Ethics Flag:** 1

**Summary:**

This paper considers sequence models based on state space models and explores various axes related to the "decay" component in these models. Specifically, the paper considers the following four axes:

1. (_Parameterization Strategy_) Considers whether the decay component parameters are fixed, trainable or input dependent.
2. (_Parameter sharing_) Whether the decay component parameters are shared with other parts of the model or not.
3. (_Decay granularity_) Using scalar vs. vector decays
4. (_Positional encoding integration_) Whether the interaction of decay component parameters with positional encoding mechanism matters.

The paper notes that state space based models can be considered as (different) instantiations of the linear attention model but ultimately only considers state space model where the _state transition matrix_ is **diagonal**. Specifically, the experiments in the papers focus on five instantiations of this class of models:

* Mamba 2
* GLA
* Hgrn 2
* LightNet
* TNL

The paper makes the following observations from its experiments:

1. For scalar decay values it should not be too close to $0$ or $1$
2. Parameter sharing cannot be used arbitrarily.
3. For same parameter settings, vector decays outperform scalar decays. But scalar decays can outperform vector delays and such decay models need higher median decay value.
4. For decay values close to $1$, the choice of positional encoding between RoPE and TPE does not seem to matter.

The paper then presents a new simple decay parameterization scheme and shows that on average it performs better on Mamba 2 on various language modeling tasks.

**Questions To Authors:**

These mirror the above five weaknesses:

1. Please clarify the reason for focusing on the diagonal state transition matrix. More importantly, please clarify how much (if any) of the takeaways from the experiments in the paper can be extended to Linear Attention (or even DPLR transition matrices)? If the answer to the latter question is not much then the scope of the writing of the paper has to be made narrow enough to cover the correct set of models that the results apply to.
2. Could you please _explicitly_ state _exactly_ which of the takeaways in Sections 5.1 to 5.4 lead to the very specific proposal in Section 5.5? Currently I do not see any specific connections. It seems the proposed simplification "came out of nowhere." It would really help if the paper explicitly points out we get to Sec 5.5 from Sections 5.1 -5.4
3. How do the results in Table 3 compare with corresponding numbers for non-state space models (these should ideally include Transfer as well other recent non-state space based Linear Attention models that get close to Transformer performance, e.g. Based)? This will give a good baseline to the reader to get a sense of how significant the improvements in this paper are.
4. Please see above.
5. How does the framework in this paper compare  with the general framework in the Mamba 2 paper?

Post-Rebuttal Comment
-----------------------------

Thanks to the authors' responses, which have addressed most of my question(s). During the process the authors clarified that the paper is _not_ proposing a new framework, which dims by excitement about the paper. At this point this  more like a $6.5$ paper for me. However since I have to put in a whole number for a rating, I'm upping my score from a $4$ to a $6$ (but leaving the point about $6.5$ in here in case the AC wants to take that into account.

Detailed Comments for Authors
-----------------------------

Here are some other minor presentation issues that I found with the paper:

* [Line 124] I think S4 should be stated as an example of DPLR as well (given that as far as I know that was the paper that first proposed this family of state transition matrices to study).
* [Table 2] Please highlight the best model (and ideally the 2nd best model) for each benchmark. It is hard to give a quick sense of so many numbers. Highlighting the best in each class will give a quick visual summary of how the models stack up against each other.
* [Line 320] I do not see how the claim on "reduced complexity" follows: please elaborate.

**Reasons To Accept:**

1. The paper considers a potentially important parameter of the upcoming state space based models and does a systematic study the various design choices.
2. The experiments that are presented seem to be fairly comprehensive would be useful for folks working on the five specific model architectures considered in this paper.

**Reasons To Reject:**

1. The actual set of experiments are much narrower than what the title, abstract and intro suggest. Ultimately the experiments are on _five_ specific models, all of which fall into the state space based models and that too only state space models that use a diagonal state transition matrix. It definitely does not include the larger class of Linear Attention models. Even within state space models, it does not consider diagonal plus low rank (DPLR) setup (which was the original setup of the S4 model [Gu, Goel and Re, ICLR 2022]). This to me diminishes the value of the "takeaways" from the experiments. E.g. do the takeaways hold for even DPLR transition matrix?
2. The paper presents a new simplified model. The paper at start of Sec 5.5 stated "Based on the previous analysis, we propose a simple decay parameterization scheme." However, I do not see why/how the specific design choices for the new scheme follow from the analysis in Section 5.1- 5.4
3. The paper presents comparison of the new proposed scheme with Mamba 2 and presents some improvements. However, I'm not sure how to interpret whether the improvement is significant or not.
4. A point related to the above: there are no comparisons in the experiments with Transformer and (non-state space based) Linear Attention models. It would help in answering the question above e.g. if the experiments results show that the new scheme closes any gaps between Mamba 2 and other Transformer and Linear Attention models.
5. I do like the fact that the paper present a general framework to think about these models but (somewhat related to the above two points), I'm curious how this framework compares to the framework in the Mamba 2 paper that also connects state space models to Linear Attention.

---

> ### Author Response · Authors · 2025-05-31
> **Response to Reviewer xr9X(Part 1)**
>
> Q1. Experiments are much narrower than what is in the title.
>
> R1. Thank you for your suggestions. First, SSM, Linear Attention, and Linear RNN are essentially different formulations of the same model.
> We list their mapping relationships in the table below, and we uniformly use Linear Attention to describe them in the paper:
>
> | Methods | Query | Key | Value | Decay |
> |---------|-------|-----|-------|-------|
> | Linear Attention | $\mathbf{q}_t$ | $\mathbf{k}_t$ | $\mathbf{v}_t$ | 1 |
> | TNL/RetNet | $\mathbf{q}_t$ | $\mathbf{k}_t$ | $\mathbf{v}_t$ | $\lambda$ |
> | RFA/xLSTM | $\mathbf{q}_t$ | $\mathbf{k}_t$ | $\mathbf{v}_t$ | $\lambda_t$ |
> | Mamba2 | $\mathbf{C}_t$ | $\mathbf{B}_t$ | $\mathbf{x}_t$ | $\mathbf{A}_t$ |
> | RWKV | $\mathbf{R}_t$ | $k_t$ | $\mathbf{v}_t$ | $\mathrm{diag}(\mathbf{w}_t)$ |
> | GLA | $\mathbf{q}t$ | $\mathbf{k}_t$ | $\mathbf{v}_t$ | $\mathrm{diag}(\gamma_t)$ |
> | HGRN2/MetaLA | $\mathbf{q}_t$ | $1-\lambda_t$ | $\mathbf{v}_t$ | $\mathrm{diag}(\lambda_t)$ |
>
> So far, the mainstream Linear Attention models still primarily use diagonal matrices for state transition, such as Mamba1/2, GLA, HGRN2, etc.
>
> On the other hand, the diagonal part of DPLR is also a type of decay. We will add discussion of this part in subsequent versions and have supplemented experiments (see R2).
>
> Q2. Whether the conclusions apply to DPLR.
>
> R2. Thank you for your suggestion. In response, we have conducted additional experiments to evaluate the performance of *Simple Decay* within the DPLR scenario, specifically comparing scalar decay and vector decay under a consistent parameterization scheme.
>
> The results indicate the following:
>
> * In DPLR settings, scalar decay consistently underperforms relative to vector decay when evaluated under the same parameterization.
> * *Simple Decay* achieves superior average perplexity (PPL) and accuracy across nearly all DPLR scenarios.
>
> These findings further support the effectiveness and robustness of *Simple Decay* and highlight the advantages of vector decay in this context.
>
> | Method | Parameters | Loss | Wikitext | Lambada_openai | AVG_PPL | BoolQ | PIQA | Hellaswag | Winogrande | Arc_easy | Arc_challenge | Openbookqa | Social_iqa | AVG_CSR |
> |--------------------|------------|--------|----------|----------------|---------|-------|-------|-----------|------------|----------|---------------|------------|------------|---------|
> | DPLR(Scalar Decay) | | | | | | | | | | | | | | |
> | Baseline | 0.1649 | 2.9645 | 40.67 | 121.84 | 81.25 | 60.1 | 64.1 | 32.7 | 50.0 | 53.6 | 25.3 | 30.8 | 36.7 | 44.2 |
> | Simple Decay | 0.1649 | 2.9408 | 39.04 | 103.70 | 71.37 | 60.1 | 64.0 | 33.0 | 50.9 | 53.8 | 25.0 | 31.0 | 38.1 | 44.5 |
> | DPLR(Vector Decay) | | | | | | | | | | | | | | |
> | Baseline | 0.1672 | 2.9372 | 39.03 | 83.40 | 61.22 | 61.0 | 63.9 | 33.7 | 50.6 | 54.9 | 25.2 | 31.4 | 38.6 | 44.9 |
> | Simple Decay | 0.1672 | 2.9202 | 37.85 | 73.46 | 55.66 | 60.1 | 64.9 | 33.8 | 48.2 | 53.6 | 25.3 | 30.8 | 36.5 | 44.1 |
> | DPLR(Scalar Decay) | | | | | | | | | | | | | | |
> | Baseline | 0.4183 | 2.736 | 30.06 | 56.07 | 43.07 | 58.4 | 67.6 | 39.4 | 51.9 | 58.4 | 27.1 | 33.6 | 36.8 | 46.6 |
> | Simple Decay | 0.4183 | 2.7173 | 29.05 | 46.00 | 37.52 | 61.1 | 67.6 | 39.5 | 51.1 | 61.2 | 29.7 | 34.0 | 38.7 | 47.9 |
> | DPLR(Vector Decay) | | | | | | | | | | | | | | |
> | Baseline | 0.4244 | 2.7317 | 29.32 | 45.21 | 37.26 | 61.0 | 67.4 | 39.7 | 50.4 | 59.6 | 29.5 | 32.6 | 37.7 | 47.2 |
> | Simple Decay | 0.4244 | 2.7192 | 28.45 | 43.25 | 35.85 | 60.7 | 67.0 | 40.0 | 50.3 | 60.3 | 27.7 | 34.6 | 38.5 | 47.4 |
> | DPLR(Scalar Decay) | | | | | | | | | | | | | | |
> | Baseline | 1.454 | 2.5233 | 23.09 | 26.56 | 24.83 | 61.5 | 70.0 | 47.1 | 53.1 | 65.4 | 33.1 | 35.4 | 40.8 | 50.8 |
> | Simple Decay | 1.454 | 2.5074 | 22.42 | 23.08 | 22.75 | 61.4 | 71.0 | 47.4 | 53.8 | 65.5 | 31.9 | 36.8 | 40.0 | 51.0 |
> | DPLR(Vector Decay) | | | | | | | | | | | | | | |
> | Baseline | 1.4658 | 2.5081 | 22.54 | 22.33 | 22.43 | 60.8 | 69.6 | 48.1 | 53.5 | 66.9 | 32.7 | 36.0 | 40.0 | 50.9 |
> | Simple Decay | 1.4658 | 2.4981 | 22.04 | 21.17 | 21.60 | 60.9 | 69.8 | 48.4 | 54.3 | 66.5 | 32.6 | 34.2 | 40.5 | 50.9 |

---

> > ### Comment · Reviewer_wY2B · 2025-06-04
> > **Thanks for the clarifications**
> >
> > Thanks for your responses: they handle most of my comments.
> >
> > Some followup questions/comments:
> >
> > * I think it would be useful to also include the Transformer baselines in Table 3 in the paper. It then makes it more apparent how significant the improvement of the _Simple Decay_ scheme is in closing the gap between Mamba 2 and Transformers.
> > * Sorry I was not clear enough when I asked for a comparison of your framework with that in Mamba 2. I was not asking Mamba 2 as a special case of your setup. In the Mamba 2 paper, the authors present a general framework that connects linear attention and SSM models, specifically their notion of State Space Duality. I was hoping for a comparison between the framework in your paper and State Space Duality.
> > * Sorry I forgot to ask this in my original review: why where the four specific axes chosen for study (and not something else e.g. the class of kernel functions on Linear Attention that maps the input to the query, key values)? Specifically, was there a systematic process by which these four axes were chosen?

---

> > > ### Author Response · Authors · 2025-06-05
> > > **Response to Reviewer wY2B**
> > >
> > > Q1. Add Transformer baseline for Table 3.
> > >
> > > R1. Thank you for your suggestion. We list the updated version of Table 3 with the Transformer Baseline added below, and will include the Transformer Baseline for Table 3 in subsequent versions:
> > > | Method | Pa | Loss | Wiki | LMB | AVG | BOQA | PIQA | Hella | Wino | ARC-E | ARC-C | OBQA | SOQA | AVG |
> > > |--------------|------|-------|--------|-------|--------|-------|-------|-------|-------|-------|-------|-------|-------|-------|
> > > | Mamba2 | 1.45 | 2.514 | 22.79 | 25.15 | 23.97 | 61.71 | 70.02 | 47.68 | 52.80 | 67.09 | 30.72 | 37.60 | 39.82 | 50.93 |
> > > | Simple Decay | 1.45 | 2.516 | 22.91 | 24.52 | 23.71 | 61.74 | 70.35 | 47.79 | 53.04 | 66.08 | 32.25 | 36.80 | 39.76 | 50.98 |
> > > | Simple Decay | 1.45 | 2.512 | 22.73 | 25.63 | 24.18 | 60.67 | 70.57 | 47.47 | 53.67 | 65.61 | 31.57 | 36.20 | 40.48 | 50.78 |
> > > | Simple Decay | 1.45 | 2.511 | 22.73 | 23.92 | 23.32 | 62.11 | 70.24 | 48.05 | 51.14 | 65.99 | 32.42 | 35.40 | 41.25 | 50.83 |
> > > | Simple Decay | 1.45 | 2.511 | 22.63 | 24.31 | 23.47 | 58.44 | 70.13 | 47.71 | 55.88 | 66.71 | 33.36 | 36.40 | 40.17 | 51.10 |
> > > | Baseline | | | | | | | | | | | | | | |
> > > | LLaMA | 1.44 | 2.520 | 22.286 | 25.07 | 23.678 | 61.68 | 69.42 | 46.89 | 53.2 | 65.82 | 30.89 | 35.4 | 39.82 | 50.39 |
> > >
> > > Q2. Comparison between the framework in this paper and State Space Duality.
> > >
> > > R2. Thank you for your suggestion. We discuss here the connection between SSD and our framework.
> > > The SSD in Mamba2 allows SSM to be trained in parallel and perform inference in RNN form. Specifically, Mamba2 shows (using Mamba2's notation):
> > >
> > > Recurrent form:
> > > $$s_t = A_t s_{t-1} + B_tx_t^\top, o_t^\top=C_t^\top s_t.$$
> > >
> > > is equivalent to the following parallel form:
> > > $$L_{ij}= \prod_{t=j+1}^i A_t, i\ge j, L_{ij}=0, i < j, \\
> > > O=[[CB^\top ]\odot L]X.$$
> > > Under Linear Attention notation, the above fact can be described as:
> > >
> > > Recurrent form:
> > > $$s_t = \lambda_t s_{t-1} + k_tv_t^\top, o_t^\top=q_t^\top s_t.$$
> > > is equivalent to the following parallel form:
> > > $$L_{ij}= \prod_{t=j+1}^i A_t, i\ge j, L_{ij}=0, \\
> > > O=[[Q K^\top ]\odot L] V.$$
> > > Our intention is not to propose a new framework. We provide this table to explain the connection between SSM and Linear Attention to inform readers that the decay conclusions we discuss here can be applied to both Linear Attention and SSM models.
> > >
> > > Q3. Why were the four specific axes chosen for study?
> > >
> > > R3. Thank you for your suggestion. Let me describe the research motivation.
> > >
> > > In the LLM scenario for Linear Attention, the most core part is the recurrent form of its state:
> > > $$s_t = s_{t-1} + k_t v_t^\top.$$
> > > However, researchers found that this recurrent form doesn't work very well, so subsequent papers mainly focus on improving this recursion. For example, TNL/RetNet introduced data-independent decay:
> > > $$s_t = \lambda s_{t-1} + k_t v_t^\top.$$
> > > Mamba2 introduced data-dependent decay based on decay, where the recursion becomes:
> > > $$s_t = \lambda_t s_{t-1} + k_t v_t^\top.$$
> > > GLA/Hgrn2 changed scalar decay to vector decay:
> > > $$s_t = \mathrm{diag}(\lambda_t) s_{t-1} + k_t v_t^\top.$$
> > > DPLR further complicated the recurrent formula:
> > > $$s_t = (\mathrm{diag}(\lambda_t) + a_t b_t^\top) s_{t-1} + k_t v_t^\top.$$
> > > As we can see, decay has become a core component of Linear Attention, but different papers use decay quite differently:
> > > - Different papers vary greatly in how they compute decay, which we call the Parameterization strategy problem;
> > > - Whether to use additional parameters to compute decay, which we call the Parameter sharing problem;
> > > - Different papers use different decay granularities, which we call the Decay granularity problem;
> > > - Whether RPE and Decay are compatible, which we call the Positional encoding integration problem;
> > >
> > > Based on the above reasons, we believe that systematically studying Decay is very valuable.
> > > Regarding the kernel function you mentioned, many papers have already discussed this issue, such as Table 6 in Mamba2, Table 8 in TNL, and Table 1 in Deltanet, so we did not discuss this here.

---

> > > > ### Comment · Reviewer_wY2B · 2025-06-08
> > > > **Thanks!**
> > > >
> > > > Thanks for the follow up clarifications. The connections to State Space Duality is not what what y'all stated in the above comment but the comment "_Our intention is **not** to propose a new framework_" (emphasis added)  makes my ask less relevant. Otoh the promise of a framework was what originally excited me but still the rebuttal has convinced me to up my score by $2$ (please see my updated review for more on the score part).

---

> ### Author Response · Authors · 2025-05-31
> **Response to Reviewer xr9X(Part 2)**
>
> Q3. Which parts the simplification is based on.
>
> R3. Thank you for your suggestion. We derived the final Simple Decay mainly based on the following analysis:
>
> - Section 5.1 analyzes the advantages and disadvantages of various parameterized decay schemes. We found that decay performs best when the median is around 0.8. Based on this, we designed simple decay to ensure that the decay median in the early stages of network training is at a value relatively close to 1;
> - Section 5.2 indicates that parameter sharing cannot be used arbitrarily, and its effect is equivalent to low rank decay, so we used low rank decay in the final Simple Decay scheme;
> - Section 5.4 indicates that in most cases, using Decay no longer requires RPE, so our Simple decay also does not use RPE;
>
> Q4. Comparison with Mamba2
>
> R4. Our goal is not to propose a decay scheme better than Mamba2, but to analyze the design of decay and provide a simpler scheme with comparable effectiveness. We list the implementation methods of Mamba Decay and Simple Decay below:
> Mamba2:
> ```
> # init
> dt = torch.exp(torch.rand(self.decay_dim,) * (math.log(dt_max) - math.log(dt_min)) + math.log(dt_min))
> dt = torch.clamp(dt, min=dt_init_floor)
> # Inverse of softplus: https://github.com/pytorch/pytorch/issues/72759
> inv_dt = dt + torch.log(-torch.expm1(-dt))  # expm1(x) = exp(x) - 1
> A = torch.empty(self.decay_dim, dtype=torch.float32).uniform_(
>     *A_init_range
> )
> self.dt_bias = nn.Parameter(inv_dt, requires_grad=True)
> self.log_A = nn.Parameter(log_A, requires_grad=True)
>
> # compute
> k = F.softplus(self.k_proj(x) + self.dt_bias)
> log_f = -self.log_A.float().exp() * k
> ```
>
> Simple Decay:
> ```
> # take x = 0 as median, 1 / (1 + exp(-(median + delta))) = a => 1 + exp(-delta) = 1 / a => exp(-delta) = (1 / a - 1) -> exp(delta) = a / (1 - a) => delta = log(a / (1 - a))
> a = self.threshold
> delta = torch.ones(self.decay_dim) * math.log(a / (1 - a))
> self.delta = nn.Parameter(delta, requires_grad=True)
>
> # compute
> log_f = F.logsigmoid(self.k_proj(x) + self.delta)
> ```
> As can be seen, Simple Decay's initialization and computation parts are more concise and easier to understand, with fewer hyperparameters, and the final effect is comparable to or better than Mamba2.

---

> ### Author Response · Authors · 2025-05-31
> **Response to Reviewer xr9X(Part 3)**
>
> Q5. Adding Transformer Baseline
>
> R5. Thank you for your suggestion. We have added the Transformer Baseline (LLaMA architecture), with results as follows:
>
> | Method | Pa | Loss | Wiki | LMB | AVG | BOQA | PIQA | Hella | Wino | ARC-E | ARC-C | OBQA | SOQA | AVG |
> |--------|--------|------|------|--------|-----|-------|------|-------|------|------|------|--------|-------|-----|
> | **Vector decay** | | | | | | | | | | | | | | |
> | Mamba2 | 1.45 | 2.514 | 22.8 | 25.1 | 24.0 | 61.7 | 70.0 | 47.7 | 52.8 | 67.1 | 30.7 | 37.6 | 39.8 | 50.9 |
> | GLA | 1.45 | 2.530 | 23.4 | 29.4 | 26.4 | 57.3 | 69.3 | 47.3 | 54.1 | 66.5 | 33.8 | 36.6 | 39.8 | 50.6 |
> | Hgrn2 | 1.45 | 2.526 | 23.2 | 24.3 | 23.8 | 59.7 | 70.1 | 47.2 | 52.5 | 65.9 | 33.5 | 35.4 | 39.5 | 50.5 |
> | LightNet | 1.45 | 2.561 | 25.2 | 34.8 | 30.0 | 58.9 | 69.4 | 43.3 | 53.8 | 64.6 | 30.6 | 34.8 | 40.1 | 49.4 |
> | **Mamba abl** | | | | | | | | | | | | | | |
> | Mamba2 wo A | 1.45 | 2.513 | 22.8 | 24.3 | 23.5 | 62.4 | 69.9 | 47.4 | 55.6 | 66.6 | 32.1 | 33.2 | 40.1 | 50.9 |
> | Mamba2 wo t | 1.45 | 2.585 | 25.3 | 31.5 | 28.4 | 58.5 | 68.9 | 44.7 | 50.8 | 64.9 | 30.2 | 36.4 | 39.8 | 49.3 |
> | Mamba2 wo A wo t | 1.45 | 2.526 | 23.4 | 25.8 | 24.6 | 61.0 | 69.6 | 47.7 | 53.2 | 67.5 | 32.4 | 38.0 | 39.5 | 51.1 |
> | **Parameter share** | | | | | | | | | | | | | | |
> | Mamba2 | 1.45 | 2.517 | 22.8 | 24.6 | 23.7 | 60.7 | 69.9 | 47.4 | 54.1 | 66.8 | 30.6 | 36.8 | 39.9 | 50.8 |
> | GLA | 1.45 | 2.583 | 25.5 | 35.9 | 30.7 | 61.7 | 69.4 | 45.5 | 50.8 | 65.5 | 30.9 | 35.0 | 39.4 | 49.8 |
> | Hgrn2 | 1.45 | 2.529 | 23.3 | 24.2 | 23.7 | 58.0 | 70.2 | 47.2 | 51.1 | 67.0 | 31.2 | 36.2 | 40.2 | 50.1 |
> | LightNet | 1.45 | 2.620 | 26.0 | 49.1 | 37.6 | 60.9 | 68.8 | 42.7 | 50.9 | 61.5 | 30.5 | 33.8 | 38.8 | 48.5 |
> | **Scalar decay** | | | | | | | | | | | | | | |
> | Mamba2 | 1.45 | 2.529 | 23.4 | 28.3 | 25.8 | 56.6 | 69.3 | 47.0 | 51.7 | 66.7 | 31.7 | 38.2 | 40.9 | 50.3 |
> | GLA | 1.45 | 2.550 | 23.8 | 28.9 | 26.3 | 60.6 | 70.0 | 46.3 | 52.6 | 65.9 | 32.7 | 35.8 | 40.1 | 50.5 |
> | Hgrn2 | 1.45 | 2.541 | 24.2 | 32.0 | 28.1 | 60.0 | 69.3 | 45.9 | 53.5 | 66.0 | 30.7 | 35.0 | 39.4 | 50.0 |
> | LightNet | 1.45 | 2.574 | 24.3 | 33.3 | 28.8 | 62.0 | 69.3 | 45.1 | 51.3 | 65.3 | 29.7 | 36.0 | 38.7 | 49.7 |
> | TNL | 1.45 | 2.552 | 24.3 | 29.4 | 26.9 | 61.3 | 69.9 | 45.9 | 53.8 | 66.6 | 30.3 | 34.8 | 40.3 | 50.4 |
> | TNL-L | 1.45 | 2.545 | 23.7 | 29.0 | 26.4 | 59.6 | 70.7 | 46.1 | 51.4 | 64.1 | 30.0 | 35.8 | 39.3 | 49.6 |
> | **RoPE** | | | | | | | | | | | | | | |
> | Mamba2 | 1.45 | 2.532 | 23.5 | 28.2 | 25.9 | 60.7 | 69.4 | 46.6 | 53.7 | 65.7 | 30.9 | 35.6 | 40.3 | 50.4 |
> | GLA | 1.45 | 2.580 | 25.5 | 35.0 | 30.2 | 60.1 | 69.0 | 45.3 | 54.2 | 65.2 | 31.6 | 35.4 | 39.1 | 50.0 |
> | Hgrn2 | 1.45 | 2.559 | 24.6 | 29.3 | 27.0 | 59.1 | 69.2 | 45.6 | 51.5 | 66.0 | 31.7 | 35.4 | 39.9 | 49.8 |
> | LightNet | 1.45 | 2.570 | 24.5 | 30.1 | 27.3 | 61.4 | 69.4 | 45.5 | 52.4 | 64.9 | 29.5 | 34.6 | 39.1 | 49.6 |
> | TNL | 1.45 | 2.547 | 24.2 | 26.7 | 25.5 | 60.9 | 70.2 | 46.1 | 53.7 | 66.1 | 31.6 | 35.4 | 39.6 | 50.4 |
> | TNL-L | 1.45 | 2.553 | 24.0 | 31.8 | 27.9 | 61.6 | 69.8 | 46.1 | 53.7 | 66.0 | 31.3 | 36.2 | 39.9 | 50.6 |
> | **Tpe** | | | | | | | | | | | | | | |
> | Mamba2 | 1.45 | 2.531 | 23.4 | 28.9 | 26.2 | 61.7 | 70.8 | 47.0 | 54.1 | 67.0 | 32.8 | 37.0 | 39.2 | 51.2 |
> | GLA | 1.45 | 2.569 | 25.1 | 36.0 | 30.5 | 61.8 | 68.8 | 45.5 | 53.2 | 65.6 | 31.2 | 36.4 | 39.5 | 50.2 |
> | Hgrn2 | 1.45 | 2.554 | 24.3 | 31.0 | 27.7 | 61.7 | 69.5 | 46.3 | 52.6 | 65.5 | 31.7 | 34.8 | 39.8 | 50.2 |
> | LightNet | 1.45 | 2.567 | 24.4 | 31.1 | 27.8 | 61.1 | 69.4 | 45.3 | 52.8 | 64.9 | 33.1 | 35.8 | 40.1 | 50.3 |
> | TNL | 1.45 | 2.556 | 24.3 | 29.6 | 27.4 | 61.1 | 70.5 | 46.2 | 52.3 | 65.9 | 31.1 | 35.4 | 40.3 | 50.6 |
> | TNL-L | 1.45 | 2.550 | 24.0 | 30.8 | 27.4 | 61.7 | 69.9 | 45.9 | 51.9 | 67.3 | 31.6 | 35.8 | 40.3 | 50.6 |
> | **Baseline** | | | | | | | | | | | | | | |
> | LLaMA | 1.44 | 2.520 | 22.3 | 25.1 | 23.7 | 61.7 | 69.4 | 46.9 | 53.2 | 65.8 | 30.9 | 35.4 | 39.8 | 50.4 |
>
> We can see that:
> LLaMA demonstrates superior performance on perplexity (PPL) metrics, and most linear attention models tend to underperform relative to LLaMA in this regard. However, on multiple-choice tasks, linear attention models achieve performance that is comparable to LLaMA.

---

> ### Author Response · Authors · 2025-05-31
> **Response to Reviewer xr9X(Part 4)**
>
> Q6. The relationship between this paper's framework and Mamba2's framework
>
> R6. Thank you for your suggestion. We provide the correspondence between several common Linear Attention models and Mamba in the table below. Specifically, Query in Linear Attention corresponds to C in SSM, Key corresponds to B, Value corresponds to X, and Decay corresponds to A.
> | Methods | Query | Key | Value | Decay |
> |---------|-------|-----|-------|-------|
> | Linear Attention | $\mathbf{q}_t$ | $\mathbf{k}_t$ | $\mathbf{v}_t$ | 1 |
> | TNL/RetNet | $\mathbf{q}_t$ | $\mathbf{k}_t$ | $\mathbf{v}_t$ | $\lambda$ |
> | RFA/xLSTM | $\mathbf{q}_t$ | $\mathbf{k}_t$ | $\mathbf{v}_t$ | $\lambda_t$ |
> | Mamba2 | $\mathbf{C}_t$ | $\mathbf{B}_t$ | $\mathbf{x}_t$ | $\mathbf{A}_t$ |
> | RWKV | $\mathbf{R}_t$ | $k_t$ | $\mathbf{v}_t$ | $\mathrm{diag}(\mathbf{w}_t)$ |
> | GLA | $\mathbf{q}t$ | $\mathbf{k}_t$ | $\mathbf{v}_t$ | $\mathrm{diag}(\gamma_t)$ |
> | HGRN2/MetaLA | $\mathbf{q}_t$ | $1-\lambda_t$ | $\mathbf{v}_t$ | $\mathrm{diag}(\lambda_t)$ |
>
> Q7. Supplementing DPLR-related literature
>
> R7. Regarding the DPLR section, we will supplement S4 as a reference;
>
> Q8. Optimizing table visualization
>
> R8. Thank you for your suggestion. We will optimize this in subsequent versions.
>
> Q9. How Simple Decay simplifies the design
>
> R9. See R3.

---

> ### Author Response · Authors · 2025-06-08
> **Looking forward to your reply**
>
> Dear reviewer, did our response address your concerns? If you have any other questions, please feel free to ask.

---

### Official Review · Reviewer_xr9X · 2025-05-11

**Rating:** 5
**Confidence:** 3
**Ethics Flag:** 1

**Summary:**

This paper presents a systematic investigation into the design space of decay mechanisms within linear attention models. The authors discuss four key dimensions: (1) parameterization strategy for computing decay values, (2) the impact of parameter sharing for decay computation, (3) decay granularity (scalar vs. vector-based decay), and (4) compatibility with relative positional encoding methods like RoPE and TPE. Through experiments on language modeling tasks, they reveal several insights: 1. effective decay parameterization strategies are confined to a specific range; 2. arbitrary parameter sharing can be detrimental; 3. vector decay generally outperforms scalar decay under similar parameterization, though scalar decay can be superior with alternative strategies; and 4. RoPE offers limited benefits to most linear attention mechanisms that already incorporate decay. Finally, the paper proposes a simplified decay parameterization, "Simple Decay," which aims to balance performance and complexity.

**Reasons To Accept:**

* The paper attempts a systematic delineation and empirical evaluation of the decay mechanism's design space in linear attention.
* The finding that optimal decay values tend to cluster within a specific range (e.g., median values around 0.8 performing well, and values too close to 0 or 1 being detrimental) offers a useful heuristic when working with these models.
* The discussion of the compatibility of decay mechanisms with RoPE and the conclusion that decay often diminishes the impact of RoPE is an interesting point for discussion in the field.
* The proposed "Simple Decay" parameterization, while related to existing methods, offers a more concise formulation that reportedly achieves competitive or even superior performance to Mamba2 under certain initializations, which could be of interest.

**Reasons To Reject:**

* The primary proposed method, "Simple Decay," is acknowledged by the authors as being quite similar to a variant of Mamba2 (Mamba2 without A). While simplification is valuable, the algorithmic novelty appears somewhat limited, positioning the contribution more as a refinement or specific configuration of existing ideas rather than a groundbreaking new mechanism.

* Several conclusions, such as vector decay generally outperforming scalar decay (under the same parameterization) or that extreme decay values (very close to 0 or 1) leading to performance degradation (e.g., attention dilution when decay is near 1), might be perceived as confirming existing intuitions or known behaviors in attention mechanisms, rather than offering entirely surprising insights.

---

> ### Author Response · Authors · 2025-05-31
> **Response to Reviewer xr9X**
>
> Q1. Limited novelty for the Simple Decay
>
> R1. Thank you for your insightful comments. In response, we would like to emphasize that the primary contribution of this work lies in the systematic exploration of the design space of decay mechanisms within linear complexity models, rather than the proposal of a novel decay strategy per se. The *Simple Decay* method is presented as an illustrative example, derived from our empirical observations. Notably, this method demonstrates both greater simplicity and improved performance compared to existing approaches.
>
> To the best of our knowledge, prior studies have not comprehensively examined this design space, particularly with respect to the following four dimensions:
>
> (1) parameterization strategies,
>
> (2) the influence of parameter sharing,
>
> (3) decay granularity, and
>
> (4) compatibility with relative positional encoding.
>
> We believe that the insights gained from this investigation offer meaningful contributions to the understanding and development of decay mechanisms in this domain.
>
> Furthermore, to assess the generalizability of *Simple Decay*, we have evaluated its performance under the DPLR framework (including both scalar and vector decay settings), as presented in the following table:
>
> | Method | Parameters | Loss | Wikitext | Lambada_openai | AVG_PPL | BoolQ | PIQA | Hellaswag | Winogrande | Arc_easy | Arc_challenge | Openbookqa | Social_iqa | AVG_CSR |
> |--------------------|------------|--------|----------|----------------|---------|-------|-------|-----------|------------|----------|---------------|------------|------------|---------|
> | DPLR(Scalar Decay) | | | | | | | | | | | | | | |
> | Baseline | 0.1649 | 2.9645 | 40.67 | 121.84 | 81.25 | 60.1 | 64.1 | 32.7 | 50.0 | 53.6 | 25.3 | 30.8 | 36.7 | 44.2 |
> | Simple Decay | 0.1649 | 2.9408 | 39.04 | 103.70 | 71.37 | 60.1 | 64.0 | 33.0 | 50.9 | 53.8 | 25.0 | 31.0 | 38.1 | 44.5 |
> | DPLR(Vector Decay) | | | | | | | | | | | | | | |
> | Baseline | 0.1672 | 2.9372 | 39.03 | 83.40 | 61.22 | 61.0 | 63.9 | 33.7 | 50.6 | 54.9 | 25.2 | 31.4 | 38.6 | 44.9 |
> | Simple Decay | 0.1672 | 2.9202 | 37.85 | 73.46 | 55.66 | 60.1 | 64.9 | 33.8 | 48.2 | 53.6 | 25.3 | 30.8 | 36.5 | 44.1 |
> | DPLR(Scalar Decay) | | | | | | | | | | | | | | |
> | Baseline | 0.4183 | 2.736 | 30.06 | 56.07 | 43.07 | 58.4 | 67.6 | 39.4 | 51.9 | 58.4 | 27.1 | 33.6 | 36.8 | 46.6 |
> | Simple Decay | 0.4183 | 2.7173 | 29.05 | 46.00 | 37.52 | 61.1 | 67.6 | 39.5 | 51.1 | 61.2 | 29.7 | 34.0 | 38.7 | 47.9 |
> | DPLR(Vector Decay) | | | | | | | | | | | | | | |
> | Baseline | 0.4244 | 2.7317 | 29.32 | 45.21 | 37.26 | 61.0 | 67.4 | 39.7 | 50.4 | 59.6 | 29.5 | 32.6 | 37.7 | 47.2 |
> | Simple Decay | 0.4244 | 2.7192 | 28.45 | 43.25 | 35.85 | 60.7 | 67.0 | 40.0 | 50.3 | 60.3 | 27.7 | 34.6 | 38.5 | 47.4 |
> | DPLR(Scalar Decay) | | | | | | | | | | | | | | |
> | Baseline | 1.454 | 2.5233 | 23.09 | 26.56 | 24.83 | 61.5 | 70.0 | 47.1 | 53.1 | 65.4 | 33.1 | 35.4 | 40.8 | 50.8 |
> | Simple Decay | 1.454 | 2.5074 | 22.42 | 23.08 | 22.75 | 61.4 | 71.0 | 47.4 | 53.8 | 65.5 | 31.9 | 36.8 | 40.0 | 51.0 |
> | DPLR(Vector Decay) | | | | | | | | | | | | | | |
> | Baseline | 1.4658 | 2.5081 | 22.54 | 22.33 | 22.43 | 60.8 | 69.6 | 48.1 | 53.5 | 66.9 | 32.7 | 36.0 | 40.0 | 50.9 |
> | Simple Decay | 1.4658 | 2.4981 | 22.04 | 21.17 | 21.60 | 60.9 | 69.8 | 48.4 | 54.3 | 66.5 | 32.6 | 34.2 | 40.5 | 50.9 |
>
> As can be seen, simple decay achieves better average PPL and accuracy in almost all DPLR scenarios, demonstrating the generalizability of our design.
>
> Q2. Several conclusions are not entirely surprising insights.
>
> R2.
>
> Thank you for your thoughtful suggestion. We would like to clarify that, under the same parameterization scheme, scalar decay generally underperforms compared to vector decay—a distinction that, to our knowledge, has not been systematically addressed in the existing literature. For instance, while *Mamba2* outperforms *Mamba1*, it employs scalar decay, whereas *Mamba1* uses vector decay. A direct comparison between the two models might therefore lead to misleading or contradictory conclusions.
>
> Importantly, the performance differences between *Mamba1* and *Mamba2* cannot be attributed solely to their respective decay strategies, as the models differ in several other aspects, such as state size. Consequently, attributing the performance gains exclusively to the choice of decay type risks confounding variables.
>
> Our work aims to fill this gap by systematically disentangling the effects of individual design choices. Through carefully controlled experiments, we investigate which factors most significantly influence decay performance and identify which architectural components may be unnecessary.

---

> ### Author Response · Authors · 2025-06-08
> **Looking forward to your reply**
>
> Dear reviewer, did our response address your concerns? If you have any other questions, please feel free to ask.

---

> ### Author Response · Authors · 2025-06-10
> **The discussion phase has less than 24 hours until the deadline, looking forward to your reply.**
>
> Dear reviewer, there are less than 24 hours left until the review deadline. If you have any questions, please let us know.

---

### Official Review · Reviewer_kU6m · 2025-05-13

**Rating:** 8
**Confidence:** 3
**Ethics Flag:** 1

**Summary:**

This paper sorts out the design space of the decay mechanism in linear attention models into four dimensions (parametrization, parameter sharing, scalar/vector decay, and compatibility with relative position encoding) and experimented the relative merit among the design choices.
Furthermore, based on the experimental result, a simple decay parametrization is proposed and evaluated.

**Reasons To Accept:**

- The paper offers valuable insights about a key component of the linear attention models, which are strong candidates of alternative to transformers

**Reasons To Reject:**

- No particular reason comes to my mind

---

### Decision · Program_Chairs · 2025-07-08

**Decision:**

Accept

**Comment:**

Summary
===
This study provides a thorough empirical investigation into the design space of attention decay in linear-attention sequence models. Various approaches are compared via perplexity on multiple datasets and accuracy on a variety of downstream tasks. It is found that (1) scalar decays of around 0.8 are best; (2) arbitrary parameter sharing is harmful to performance; (3) vector decays outperform scalar decays under identical settings (though with some extra tuning, scalar decays can outperform vector decays); and (4) the choice of positional encoding does not significantly matter with decay ≈ 1. Using these insights, Simple Decay is proposed, and is argued to achieve the best of all worlds.

Reasons to Accept
===
* Thorough and convincing experimentation. Good variety of model architectures, model scales, and task settings. Most experiments have clear conclusions.
* Dives quite deep into the effects of an often-underappreciated parameter.
* The proposed method is empirically principled (though theoretically not well understood) performs well.

Reasons to Reject
===
* Discussions with xr9X and wY2B reveal that the presentation of the proposed method should be improved. There seemed to be a lot of feedback that was well addressed by the authors during the rebuttal period, but which nonetheless show that some clarification is needed.
* I agree with Reviewer R6Me that it's not clear why these results turned out the way they did. Understanding this would allow us to draw more general conclusions about when certain classes of methods are more likely to outperform others. This paper is already full of empirical insights, however, so I think it's reasonable to say that there wouldn't be enough room to discuss this convincingly.

[Automatically added comment] At least one review was discounted during the decision process due to quality]